# Bandit Learning in Many-to-one Matching Markets with Uniqueness Conditions

## Abstract

An emerging line of research is dedicated to the problem of one-to-one matching markets with bandits, where the preference of one side is unknown and thus we need to match while learning the preference through multiple rounds of interaction. However, in many real-world applications such as online recruitment platform for short-term workers, one side of the market can select more than one participant from the other side, which motivates the study of the many-to-one matching problem. Moreover, the existence of a unique stable matching is crucial to the competitive equilibrium of the market. In this paper, we first introduce a more general new $\tilde{\alpha}$-condition to guarantee the uniqueness of stable matching in many-to-one matching problems, which generalizes some established uniqueness conditions such as *SPC* and *Serial Dictatorship*, and recovers the known $\alpha$-condition if the problem is reduced to one-to-one matching. Under this new condition, we design an MO-UCB-D4 algorithm with $O\left(\frac{NK \log(T)}{\Delta^2}\right)$ regret bound, where $T$ is the time horizon, $N$ is the number of agents, $K$ is the number of arms, and $\Delta$ is the minimum reward gap. Extensive experiments show that our algorithm achieves uniform good performances under different uniqueness conditions.

## 1 Introduction

The data-driven matching market is faced with the problems of learning customer preference and matching the demand side with the supply side of the market to maximize the benefits of both sides. Online platforms, like Lyft, Thumbtack and Taskrabbit, make decisions for customers and service providers to match, on the basis of their diversified needs, which is abstracted as a matching market with an agent side and an arm side, and each side has a preference profile over the opposite side. They choose from the other side according to preference and perform a matching. Specific examples like pool riding in ride-share system that matches a driver to multiple riders, Slate ranking in recommender systems that a user is matched to various content at a single request Ie et al. (2019). The stability of the matching result is a key property of the market Roth & Sotomayor (1992); Abizada (2016); Park (2017).

This work takes online short-term recruitment as the main example, combine the traditional matching problem Bade (2020); Bogomolnaia & Moulin (2001); Roth & Sotomayor (1992) with the online system Gunn et al. (2022); Malgonde et al. (2020); Johari et al. (2021). Companies with short-term needs accommodate workers who are voluntarily looking for flexible probation periods. The worker preferences may be unknown in advance, thus matching while learning the preferences is necessary. The multi-armed bandit (MAB) Thompson (1933); Garivier et al. (2016); Auer et al. (2002) is an important tool for $N$ independent agents in matching market simultaneously selecting arms adaptively from received rewards at each round. And the upper confidence bound algorithm (UCB) Auer et al. (2002) is a typical MAB algorithm, which sets a confidence interval to represent uncertainty. The idea of applying MAB to one-to-one matching problems, introduced by Liu et al. (2020a), assumes that there is a central platform to make decisions for all agents. Following this, other works Liu et al. (2020b); Sankararaman et al. (2021); Basu et al. (2021) consider a more general decentralized setting without a central platform to arrange matchings, and our work is also based on this setting.

However, it is not enough to just study the one-to-one setting. In online short-term worker employment problem, employers have numerous similar short-term tasks to be recruited and workers can only

choose one task according to the company's needs at a time while one company can accept more than one employee. Each company makes a fixed ranking for candidates according to its own requirements but workers have no knowledge of companies' preferences. The reward for workers is a comprehensive consideration of salary and job environment. The online matching is in an iterative way that tasks are short-term, or if an agent do not get an ideal job, he will leave the platform or start a new competition to select another company. We abstract companies as arms and workers as agents. Each arm has a *capacity* $q$ which is the maximum number of agents this arm can accommodate. When an arm faces multiple choices, it accepts its most $q$ preferred agents. Agents thus competing for arms and may receive zero reward if losing the conflict. It is worth mentioning that arms with capacity $q$ in the many-to-one matching can not just be replaced by $q$ independent replicates with the same preference since there would be implicit competition. In addition, when multiple agents select one arm at a time, collision is unavoidable, which hinder the communication among different agents under the decentralized assumption. They cannot distinguish who is more preferred by this arm in one round as it can accept more than one agent while this can be done in one-to-one case. Communication here lets each agent learn more about preferences of arms and other agents, so as to formulate better policies to reduce collisions and learn faster about their stable results.

This work focuses on a many-to-one market under uniqueness conditions. Previous work Clark (2006); Gutin et al. (2021) emphasize the importance of constructing a unique stable matching for the equilibrium of matching problems and some existing uniqueness conditions are studied in many-to-one matching, such as *Sequential Preference Condition (SPC)* and *Acyclicity* Niederle & Yariv (2009); Akahoshi (2014). Our work is motivated by Basu et al. (2021), but the unique one-to-one mapping between arms and agents in their study which gives a surrogate threshold for arm elimination does not work in the many-to-one setting. And the uniqueness conditions in many-to-one matching are not well-studied, which also brings a challenge to identify and leverage the relationship between the resulting stable matching and preferences of two sides in the design of bandit algorithms. We propose an $\tilde{\alpha}$-condition that can guarantee a unique stable matching and recover $\alpha$-condition Karpov (2019) if reduced to the one-to-one setting. We establish the relationships between our new $\tilde{\alpha}$-condition and existing uniqueness conditions in many-to-one setting.

For clarity, in this paper, we study the bandit algorithm for a decentralized many-to-one matching market with uniqueness conditions. Under our newly proposed uniqueness condition, $\tilde{\alpha}$-*condition*, we design an MO-UCB-D4 algorithm with arm elimination to construct a stable matching result. The regret of our algorithm can be upper bounded by $O\left(\frac{NK\log(T)}{\Delta^2}\right)$, where $N$ is the number of agents, $K$ is the number of arms, and $\Delta$ is the minimum reward gap, and the regret reaches the lower bound in terms of $T$ and $\Delta$. Finally, we conduct a series of experiments to simulate our algorithm under various conditions of *Serial dictatorship*, *SPC* and $\tilde{\alpha}$-*condition* to study the stability and regret of the algorithm.

## 2 SETTING

This paper considers a many-to-one matching market $\mathcal{M} = (\mathcal{K}, \mathcal{J}, \mathcal{P})$, where $\mathcal{K} = [K]$, is a finite arm set and $\mathcal{J} = [N]$ is a finite agent set. Each arm $k$ has a fixed capacity $q_k \geq 1$. To guarantee that no agents will be unmatched, we focus on the market with $N \leq \sum_{i=1}^{K} q_i$. $\mathcal{P}$ is the fixed preference order of agents and arms, which is ranked by the mean reward. We assume that arm preference is over individuals Roth & Sotomayor (1992); Sethuraman et al. (2006); Altinok (2019), and arm preferences for agents are unknown and needed to be learned. If agent $j$ prefers arm $k$ than $k'$, i.e., $\mu_{j,k} > \mu_{j,k'}$, we denote by $k \succ_j k'$. And the preference is strict that $\mu_{j,k} \neq \mu_{j,k'}$ if $k \neq k'$. Similarly, each arm $k$ has preferences $\succ_k$ over all agents, and specially, $j \succ_k j'$ means that arm $k$ prefers agent $j$ over $j'$. Throughout, we focus on the market where all agent-arm pairs are *mutually acceptable*, that is, $j \succ_k \emptyset$ and $k \succ_j \emptyset$ for all $k \in [K]$ and $j \in [N]$.

Let a mapping $m$ be the matching result. $m_t(j)$ is the matched arm for agent $j$ at time $t$, and $\gamma_t(k)$ is the agent set matched with arm $k$[1]. At each time agent $j$ selects an arm $I_t(j)$, and we use $M_t(j)$ to denote whether $j$ is successfully matched with its selected arm. $M_t(j) = 1$ if agent $j$ is matched with $I_t(j)$, and $M_t(j) = 0$, otherwise. If multiple agents select arm $k$ at the same time, only top $q_k$ agents can successfully match. The agent $j$ matched with arm $k$ can observe the reward $X_{j,m_t(j)}(t)$, where

---

[1]The mapping $m$ is not reversible as it is not a injective, thus we do not use $m_t^{-1}(k)$.

the random reward $X_{j,k}(t) \in [0,1]$ is independently drawn from a fixed distribution with mean $\mu_{j,k}$. While the unmatched one has collisions and receives zero reward. Generally, the reward obtained by agent $j$ is $X_{j,I_t(j)}(t)\,M_t(j)$.

We say an agent $j$ and an arm $k$ form a *blocking pair* for a matching $m$ if they prefer each other over their current assignments, i.e. $k \succ_j m(j)$ and $\exists j' \in \gamma(k), j \succ_k j'$. We say a matching satisfies individually rational (IR), if $a_j \succ_{p_i} \emptyset$ and $p_i \succ_{a_j} \emptyset$ for all $i \in [N]$ and $j \in [K]$, that is, every worker prefers to find a job rather than do nothing, and every company also wants to recruit workers rather than not recruit anyone. Under the IR condition, a matching in the many-to-one setting is *stable* if there does not exist a blocking pair Salonen & Salonen (2018); Sethuraman et al. (2006).

This paper considers the matching markets under the uniqueness condition. Thus the overall goal is to find the unique stable matching between the agent side and arm side through iterations. Let $m^*(j)$ be the stable matched arm for agent $j$ under the stable matching $m^*$. The reward obtained by agent $j$ is compared against the reward received by matching with $m^*(j)$ at each time. We aim to minimize the expected stable regret for agent $j$ over time horizon $T$, which is defined as

$$R_j(T) = T\mu_{j,m^*(j)} - \mathbb{E}\left[\sum_{t=1}^{T} M_t(j)X_{j,I_t(j)}(t)\right].$$

## 3 ALGORITHM

In this section, we introduce our MO-UCB-D4 Algorithm (Many-to-one UCB with Decentralized Dominated arms Deletion and Local Deletion Algorithm) (Algorithm 1) for the decentralized many-to-one market, where there is no platform to arrange actions for agents. The MO-UCB-D4 algorithm sets multiple phases, and each phase $i$ mainly includes regret minimization block (line 6 - 12) and communication block (line 13 - 16) with duration $2^{i-1}, i = 1, 2, \cdots$.

---

**Algorithm 1** MO-UCB-D4 algorithm (for agent $j$)

**Input:**
    $\theta \in (0, 1/K), \alpha > 1$.
1: Set global dominated set $G_j[0] = \phi$
2: **for** phase $i = 1, 2, ...$ **do**
3:    Reset the collision set $C_{j,k}[i] = 0, \forall k \in [K]$;
4:    Reset active arms set $\mathtt{Ch}_j[i] = [K] \backslash G_j[i-1]$;
5:    **if** $t < 2^i + NK(i-1)$ **then**
6:        Local deletion $L_j[i] = \{k : C_{jk}[i] \geq \lceil \theta 2^i \rceil\}$;
7:        Play arm $I_t(j) \in \underset{k \in \mathtt{Ch}_j[i] \backslash L_j[i]}{\arg\max} \left( \hat{\mu}_{j,k}(t-1) + \sqrt{\frac{2\alpha \log(t)}{N_{j,k}(t-1)}} \right)$;
8:        **if** $k = I_t(j)$ is successfully matched with agent $j$, i.e. $m_t(j) = k$ **then**
9:            Update estimate $\hat{\mu}_{j,k}(t)$ and matching count $N_{j,k}(t)$;
10:      **else**
11:        $C_{j,k}[i] = C_{j,k}[i] + 1$;
12:      **end if**
13:    **else if** $t = 2^i + NK(i-1)$ **then**
14:      $\mathcal{O}_j[i] \leftarrow$ most matched arm in phase $i$;
15:      $G_j[i] \leftarrow COMMUNICATION(i, \mathcal{O}_j[i])$;
16:    **end if**
17: **end for**

---

For each agent $j$ in phase $i$, the algorithm adds arm deletion process to reduce potential conflicts, which contains global deletion and local deletion. The former eliminates the arms most preferred by agents who rank higher than agent $j$ and obtains active set $\mathtt{Ch}_j[i]$ (line 4), and the latter deletes the arms that still have many conflicts with agent $j$ after global deletion (line 6). We set a collision counter $C_{j,k}[i]$ to record the number of collisions for agent $j$ pulling arm $k$ in phase $i$.

In the regret minimization block of phase $i$, we use $L_j[i] = \{k : C_{j,k}[i] \geq \lceil \theta 2^i \rceil\}$ to represent the arms that collide more times than a threshold $\lceil \theta 2^i \rceil$ when matching with agent $j$. Arms in $L_j[i]$ are

first locally deleted to reduce potential collisions for agent $j$ (line 6). After that, agent $j$ selects an optimal action $I_t(j)$ from remaining arms in $\text{Ch}_j[i] \backslash L_j[i]$ in phase $i$ according to UCB index, which is computed by $\hat{\mu}_{j,k}(t-1) + \sqrt{\frac{2\alpha \log(t)}{N_{j,k}(t-1)}}$ (line 7), where $N_{j,k}(t-1)$ is the number that agent $j$ and arm $k$ have been matched at time $t-1$. If the selected arm is successfully matched with agent $j$, then the algorithm updates estimated reward $\hat{\mu}_{j,k}(t) = \frac{1}{N_{j,k}(t)} \sum_{s=1}^{t} 1\{I_s(j) = k \text{ and } M_s(j) = 1\} X_{j,k}(t)$ and $N_{j,k}(t)$ (line 9). Otherwise, the collision happens (line 11) and agent $j$ receives zero reward. The regret minimization block identifies the most played arm $\mathcal{O}_j[i]$ for agent $j$ in each phase $i$, which is estimated as the best arm for agent $j$, thus making optimal policy to minimize expected regret.

---

**Algorithm 2** COMMUNICATION

**Input:**
    Phase number $i$, and most played arms $\mathcal{O}_j[i]$ for agent $j$, $\forall j \in [N]$ .
1: Set $\mathcal{C} = \emptyset$;
2: **for** $t = 1, 2, \cdots, NK - 1$ **do**
3:     **if** $K(j-1) \leq t \leq Kj - 1$ **then**
4:         Agent $j$ plays arm $I_t(j) = (t \mod K) + 1$;
5:         **if** Collision Occurs **then**
6:             $\mathcal{C} = \mathcal{C} \cup \{I_t(j)\}$;
7:         **end if**
8:     **else**
9:         Play arm $I_t(j) = \mathcal{O}_j[i]$;
10:     **end if**
11: **end for**
12: RETURN $\mathcal{C}$;

---

In the communication block (Algorithm 2), there are $N$ sub-blocks, each with duration $K$. In the $\ell - th$ sub-block, only agent $\ell$ pulls arm 1, arm 2, $\cdots$, arm $K$ in round-robin while other agents select their most preferred arms estimated as the most played ones (line 4). This block aims to detect globally dominated arms for each agent $j$: $G_j[i] \subset \{\mathcal{O}_{j'}[i] : j' \succ_{\mathcal{O}_{j'}[i]} j\}$. Under the stable matching $m^*$, the globally dominated arm set for agent $j$ is denoted as $G_j^*$. After the communication block in phase $i$, each agent $j$ updates its active arm set $\text{Ch}_j[i+1]$ for phase $i+1$, by globally deleting arm set $G_j[i]$, and enters into the next phase (line 4 in Algorithm 1).

Hence, multi-phases setting can guarantee that the active sets in different phases have no inclusion relationship so that if an agent deletes an arm in a certain phase, this arm can still be selected in the later rounds. This ensures that each agent will not permanently eliminate its stable matched arm, and if agent $j$ mistakenly deletes an arm, it will not lead to linear regret.

## 4 RESULTS

### 4.1 UNIQUENESS CONDITIONS

#### 4.1.1 $\tilde{\alpha}$-CONDITION

When the preferences of agents and arms are given by some utility functions instead of random preferences, like payments for workers in the labor markets, the stable matching is usually unique. Thus the assumption of the unique stable matching is quite common in real applications. And some uniqueness conditions have important properties like consistency, which states that any stable pair leaving the market does not affect the remaining to form a stable matching. In dynamic markets where agents and arms come and go, the consistency property is desirable to keep the matching majority static Basu et al. (2021). And in this way, the market is divided into pairs with priority, which is divided into hierarchical structures, so that the design of the algorithm is inductive, and the regret is constrained to the number of sub-optimal matchings (Appendix 3). Besides, when the stable matching is unique, there would be no dispute about adopting stable matching preferred by which side, thus is fairer to both sides Cen & Shah (2022). Note that the outcome of the GS algorithm would prefer the proposal side and would be unfair to the other side Clark (2006).

In this section, we propose a new uniqueness condition, $\tilde{\alpha}$-condition. First, we introduce *uniqueness consistency (Unqc)* Karpov (2019), which guarantees robustness and uniqueness of markets.

**Definition 1.** *A preference profile satisfies uniqueness consistency if and only if*

*(i) there exists a unique stable matching $m^*$;*

*(ii) for any subset of arms or agents, the preference profile on this subset with their stable-matched pair can induce a unique stable matching.*

It guarantees that even if an arbitrary subset of stable pairs are deleted out of the system, there still exists a unique stable matching among the remaining agents and arms. This condition allows the algorithm to find the unique stable matching by detecting the stable matching pairs iteratively. To obtain the unique stable matching in the many-to-one market, we propose a new $\tilde{\alpha}$-*condition*, which is a sufficient and necessary condition for Unqc (proved in Appendix C).

We considers a finite set of arms $[K] = \{1, 2, \cdots, K\}$ and a finite set of agents $[N] = \{1, 2, \cdots, N\}$ with preference profile $\mathcal{P}$. Assume that $[N]_r = \{A_1, A_2, \cdots, A_N\}$ is a permutation of $\{1, 2, \cdots, N\}$ and $[K]_r = \{c_1, c_2, \cdots, c_K\}$ is a permutation of $\{1, 2, \cdots, K\}$. Denote $[N], [K]$ as the left order and $[N]_r, [K]_r$ as the right order. The $k$-th arm in the right order set $[K]_r$ has the index $c_k$ in the left order set $[K]$ and the $j$-th agent in the right order set $[N]_r$ has the index $A_j$ in the left order set $[N]$. Considering arm capacity, we denote $\gamma^*(c_k)$ (right order) as the stable matched agent set for arm $c_k$.

**Definition 2.** *A many-to-one matching market satisfies the $\tilde{\alpha}$-condition if,*

*(i) The left order of agents and arms satisfies*

$$\forall j \in [N], \forall k > j, k \in [K], \mu_{j, m^*(j)} > \mu_{j, k},$$

*where $m^*(j)$ is agent $j$'s stable matched arm;*

*(ii) The right order of agents and arms satisfies*

$$\forall k < k' \leq K, c_k \in [K]_r, A_{k'} \subset [N]_r, \gamma^*(c_k) \succ_{c_k} A_{\sum_{i=1}^{k'-1} q_{c_i}+1},$$

*where the set $\gamma^*(c_k)$ is more preferred than $A_{\sum_{i=1}^{k'-1} q_{c_i}+1}$ means that the least preferred agent in $\gamma^*(c_k)$ for $c_k$ is better than $A_{\sum_{i=1}^{k'-1} q_{c_i}+1}$ for $c_k$.*

Under our $\tilde{\alpha}$-*condition*, the left order and the right order satisfy the following rule. The left order gives rankings according to agents' preferences. The first agent in the left order set $[N]$ prefers arm 1 in $[K]$ most and has it as the stable matched arm. Similar properties for the agent 2 to $q_1$ since arm 1 has $q_1$ capacity. Then the $(q_1 + 1)$-th agent in the left order set $[N]$ has arm 2 in $[K]$ as her stable matched arm and prefers arm 2 most except arm 1. The remaining agents follow similarly. Similarly, the right order gives rankings according to arms' preferences. The first arm 1 in the right order set $[K]_r$ most prefers the first $q_{c_1}$ agents in the right order set $[N]_r$ and takes them as its stable matched agents. The remaining arms follow similarly.

This condition is more general than existing *SPC* condition Reny (2021) and can recover the known $\alpha$-condition in one-to-one matching market Karpov (2019). The relationship between existing uniqueness conditions and our proposed conditions will be analyzed in detail later in Section 4.1.2.

The main idea from one-to-one to many-to-one analysis is to replace individuals with sets. In general, under $\tilde{\alpha}$-*condition*, the left order satisfies that when arm 1 to arm $k - 1$ are removed, agents $\left(\sum_{i=1}^{k-1} q_i + 1\right)$ to $\left(\sum_{i=1}^{k} q_i\right)$ prefer $k$ most, and the right order means that when $A_1$ to agents $A_{\sum_{i=1}^{k-1} q_i}$ are removed, arm $k$ prefers agents $\mathcal{A}_k = \{A_{\sum_{i=1}^{k-1} q_{c_i}+1}, A_{\sum_{i=1}^{k-1} q_{c_i}+2}, \cdots, A_{\sum_{i=1}^{k} q_{c_i}}\}$, where $\mathcal{A}_k$ is the agent set that are most $q_k$ preferred by arm $k$ among those who have not been matched by arm $1, 2, \cdots, k - 1$. The $\tilde{\alpha}$-condition can be detected as follows: After running GS algorithm and finding a stable matching, we can find two orders of arms and agents by sequential elimination higher ranked agents or arms with their matching pairs. And the $\tilde{\alpha}$-condition satisfied if the two orders are identical. The next theorem gives a summary.

**Theorem 1.** *If a market $\mathcal{M} = (\mathcal{K}, \mathcal{J}, \mathcal{P})$ satisfies $\tilde{\alpha}$-condition, then $m^*(\sum_{i=1}^{j-1} q_i + 1) = m^*(\sum_{i=1}^{j-1} q_i + 2) = \cdots = m^*(\sum_{i=1}^{j} q_i) = j$ (the left order), $\gamma^*(c_k) = \mathcal{A}_k$ and $m^*(\mathcal{A}_j) = c_j$ (the right order) under stable matching.*

Under $\tilde{\alpha}$-condition, the stable matched arm may not be the most preferred one for each agent $j$, $j \in [N]$, thus (i) we do not have $m^*(j)$ to be dominated only by the agent 1 to agent $j-1$, i.e. there may exist $j' > j$, s.t. $j' \succ_{m^*(j)} j$; (ii) the left order may not be identical to the right order, we define a mapping $lr$ to match the index of an agent in the left order with the index in the right order, i.e. $A_{lr(j)} = j$. From Theorem 1, the stable matched set for arm $k$ is its first $q_k$ preferred agents $\gamma^*(c_k) = \mathcal{A}_k$. We define $lr$ as $lr(i) = \max\{j : A_j \in \gamma^*(m^*(i)), j \in [N]\}$, that is, in the right order, the mapping for arm $k \in [K]$ is the least preferred one among its most $q_k$ preferred agents. Note that this mapping is not an injective, i.e. $\exists j, j'$, s.t. agent $j = A_{lr(j)} = A_{lr(j')}$. An intuitive representation can be seen in Figure 4 in Appendix B.1.

### 4.1.2 UNIQUE STABLE CONDITIONS IN MANY-TO-ONE MATCHING

Uniqueness consistency (Unqc) leads the stable matching to a robust one which is a desirable property in large dynamic markets with constant individual departure Basu et al. (2021). A precondition of Unqc is to ensure global unique stability, hence finding uniqueness conditions is essential.

The existing unique stable conditions are well established in one-to-one setting (analysis can be found in Appendix C), and in this section, we focus on the uniqueness conditions in many-to-one market, such as *SPC*, Reny (2021), *Aligned Preference*, *Serial Dictatorship Top-top match* and *Acyclicity* Niederle & Yariv (2009); Akahoshi (2014); Reny (2021) (Definition 9, 7, 8, 10 in Appendix C.2). Akahoshi (2014) proposes a necessary and sufficient condition for unique stable matching in many-to-one matching where unacceptable agents and arms may exist on both sides. We denote this condition as *Acyclicity*$^*$. Under our setting, both two sides are acceptable, and we first give the proof that *Acyclicity*$^*$ is a necessary and sufficient condition for uniqueness in this setting (Section C.2.4). We then give relationships between our newly $\tilde{\alpha}$-condition and other existing uniqueness conditions, intuitively expressed in Figure 1, and we give proof for this section in Appendix C.2.

**Lemma 1.** *In a many-to-one matching market $\mathcal{M} = (\mathcal{K}, \mathcal{J}, \mathcal{P})$, both Serial Dictatorship and Aligned Preference can produce a unique stable matching and they are equivalent.*

**Theorem 2.** *In a many-to-one matching market $\mathcal{M} = (\mathcal{K}, \mathcal{J}, \mathcal{P})$, our $\tilde{\alpha}$-condition satisfies:*

*(i) SPC is a sufficient condition to $\tilde{\alpha}$-condition;*

*(ii) $\tilde{\alpha}$-condition is a necessary and sufficient condition to Unqc;*

*(iii) $\tilde{\alpha}$-condition is a sufficient but not necessary condition to Acyclicity$^*$.*

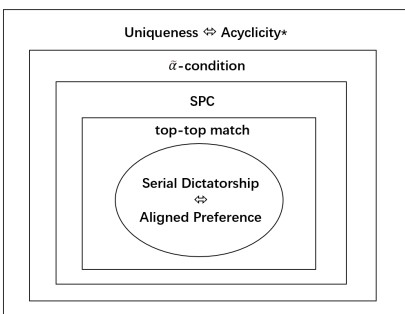

Figure 1: Relations of Uniqueness Conditions in Many-to-one Market.

### 4.2 THEORETICAL RESULTS OF REGRET

We then provide theoretical results of MO-UCB-D4 algorithm under our $\tilde{\alpha}$-condition. Recall that $G_j^*$ is the globally dominated arms for agent $j$ under stable matching $m^*$. For each arm $k \notin G_j^*$, we give the definition of the *blocking agents* for arm $k$ and agent $j$: $\mathcal{B}_{jk} = \{j' : j' \succ_k j, k \notin G_j^*\}$, which contains agents more preferred by arm $k$ than $j$. The *hidden arms* for agent $j$ is $\mathcal{H}_j = \{k : k \notin G_j^*\} \cap \{k : \mathcal{B}_{jk} \neq \emptyset\}$. The reward gap for agent $j$ and arm $k$ is defined as $\Delta_{jk} = |\mu_{j,m^*(j)} - \mu_{j,k}|$ and the minimum reward gap across all arms and agents is $\Delta = \min_{j \in [N]}\{\min_{k \in [K]} \Delta_{j,k}\}$. We assume that the reward is different for each agent, thus $\Delta_{j,k} > 0$ for every agent $j$ and arm $k$.

**Theorem 3.** *(Regret upper bound) Let $J_{\max}(j) = \max\{j + 1, \{j' : \exists k \in \mathcal{H}_j, j' \in \mathcal{B}_{jk}\}\}$ be the max blocking agent for agent $j$ and $f_{\tilde{\alpha}}(j) = j + lr_{\max}(j)$ is a fixed factor depends on both the left order and the right order for agent $j$. Following MO-UCB-D4 algorithm with horizon $T$, the expected regret of a stable matching under $\tilde{\alpha}$-condition (Definition 2) for agent $j \in [N]$ is upper bounded by*

$$\mathbb{E}\left[R_j(T)\right] \leq \sum_{k \notin G_j^* \cup m^*(j)} \frac{8\alpha}{\Delta_{jk}} \left(\log(T) + \sqrt{\frac{\pi}{\alpha} \log(T)}\right) + \sum_{k \notin G_j^*} \sum_{j' \in \mathcal{B}_{jk}: k \notin G_{j'}^*} \frac{8\alpha\mu_{j,m^*(j)}}{\Delta_{j'k}^2} \left(\log(T) + \sqrt{\frac{\pi}{\alpha} \log(T)}\right)$$

$$+ c_j \log_2(T) + O\left(\frac{N^2 K^2}{\Delta^2} + \left(\min(1, \theta|\mathcal{H}_j|) f_{\tilde{\alpha}}(J_{\max}(j)) + f_{\tilde{\alpha}}(j) - 1\right) 2^{i^*} + N^2 K i^*\right),$$

*where $i^* = \max\{8, i_1, i_2\}$ ($i_1, i_2$ are defined in equation 3), and $lr_{\max}(j) = \max\{lr(j') : 1 \leq j' \leq j\}$, is the maximum right order mapping for agent $j'$ who ranks higher than $j$.*

From Theorem 3, the scale of the regret upper bound under $\tilde{\alpha}$-*condition* is $O\left(\frac{NK \log(T)}{\Delta^2}\right)$.

**Proof Sketch of Theorem** 3. The main proof idea is how agents settle down to their stable matched arms inductively. Agent 1 will find its stable matched arm 1 at first since arm 1 is the most preferred arm for agent 1. The same is true for agent 2 to agent $q_1$. When they all settle down with arm 1, then agent $q_1 + 1$ will find its stable arm 2 since agent $q_1 + 1$ has deleted arm 1 in the communication block and thus arm 2 becomes its most preferred arm. We can show by induction that agent $j$ will find its stable matched arm after agent 1 to $j - 1$ has settled down. The regret of agent $j$ can be decomposed into four parts: sub-optimal play, collision, communication, and local deletion. Both collisions between agent $j$ and other agents in the blocking agent set and sub-optimal play are due to the wrong estimation of UCB index (Lemma 6). Communication regret can be bounded by the length of the communication block. Local deletion regret can be controlled by the threshold we set (line 6 in Algorithm 1). The regret bound is decomposed as follows, and the complete proof can be seen in Section 3.

**Lemma 2.** *(Regret Decomposition) For a stable matching under $\tilde{\alpha}$-condition, the upper bound of regret for the agent $j \in [N]$ under our algorithm can be decomposed by:*

$$\mathbb{E}\left[R_j(T)\right] \leq \underbrace{\mathbb{E}\left[S_{F_{\alpha j}}\right]}_{\text{(Regret before phase } F_{\alpha j})} + \underbrace{\min(\theta|\mathcal{H}_j|, 1)\mathbb{E}\left[S_{V_{\alpha j}}\right]}_{\text{(Local deletion)}} + \underbrace{\left((K - 1 + |\mathcal{B}_{j,m^*(j)}|)\log_2(T) + NK\mathbb{E}\left[V_{\alpha j}\right]\right)}_{\text{(Communication)}}$$

$$+ \underbrace{\sum_{k \notin G_j^*} \sum_{j' \in \mathcal{B}_{j,k}: k \notin G_{j'}^*} \frac{8\alpha\mu_{j,m^*(j)}}{\Delta_{j',k}^2} \left(\log(T) + \sqrt{\frac{\pi}{\alpha} \log(T)}\right)}_{\text{(Collision)}}$$

$$+ \underbrace{\sum_{k \notin G_j^* \cup m^*(j)} \frac{8\alpha}{\Delta_{j,k}} (\log(T) + \sqrt{\frac{\pi}{\alpha} \log(T)})}_{\text{(Sub-optimal play)}} + NK\left(1 + (\phi(\alpha) + 1)\frac{8\alpha}{\Delta^2}\right),$$

*where $F_{\alpha j}, V_{\alpha j}$ are the time points when agent $j$ enters into $\tilde{\alpha}$-Good phase and $\tilde{\alpha}$-Low Collision phase respectively, are defined in Appendix B.2.*

## 5 DIFFICULTIES AND SOLUTIONS

**From one-to-one setting to many-to-one setting** First, arm preference is difficult to learn in a decentralized many-to-one setting. Influenced by capacity, in communication block, when two agents select one arm at a time, as an arm can accept more than one agent, these two cannot distinguish who is more preferred by this arm, while it can be done in one-to-one markets. Thus identifying arm preference for each agent encounters more challenges, and then influences total regret. In order to solve this, we introduce the dominated arm set $G_j^*$ into communication block to identify arms who are preferred by higher ranked agents than agent $j$. The arm set $G_j^*$ is one of the main sources that prevent agent $j$ from forming a stable matching, and it will be deleted before each phase to reduce collisions.

Second, the idea from one-to-one to many-to-one is a transition from individual to set. It is natural to split sets into individuals or correspond sets to individuals. Although we assume that arm preference is over individuals Roth & Sotomayor (1992); Sethuraman et al. (2006); Altinok (2019), the agents matched by one arm are not independent. Specially, arms with capacity $q$ can not just be replaced by $q$ independent individuals with the same preference. Since there would be implicit competition among different replicates of one arm, and he can reject previously accepted agents when he faces a more preferred agent. In addition, considering capacity, the matching result for each arm $k$ is a set rather than an individual. In order to give a description of a uniqueness condition, we need to give a threshold for the range of stable matched agents set. The $lr$ in Basu et al. (2021) is a one-to-one mapping that corresponds the agent index in the left order and the agent index in the right order, which is related to regret bound (Theorem 3 in Basu et al. (2021) and Theorem 3 in our work). While it does not hold in our setting. We construct a new mapping $lr$ (Figure 4 in Appendix B) which connects the index of agents in two orders in many-to-one setting. $lr$ maps each arm $k$ to the least preferred one of its stable matched agents in the right order, thus giving a mapping between individuals and individuals.

**From $\alpha$-condition to $\tilde{\alpha}$-condition** In general markets, preferences are difficult to learn when one arm can accommodate multiple agents. We consider the market with uniqueness condition. For one thing, equilibrium plays an important role in the fairness and stability of matching problems. For another, to reduce the conflicts among agents, we adopt an arm deletion idea and Unqc (Definition 1) can ensure that the deletion does not affect the stable matching.

Our work extends $\alpha$-condition to many-to-one setting, which needs to define preferences among sets. However, there might be an exponential number of sets due to the combinatorial structure and simply constraining preferences over all possible sets will lead to high complexity. Motivated by $\alpha$-condition which characterizes properties of matched pairs in one-to-one setting, we come up with a possible constraint by regarding the arm and the least preferred agent in its matched set as the *matched pair* and define preferences according to this grouping. It turns out that we only need to define arm preferences over disjoint agent sets to complete this extension as $\alpha$-condition is defined under the stable matching, which can also fit the regret analysis well. Under this $\tilde{\alpha}$-condition, it induces a hierarchy in the matching market, which reduces the regret bound from collision block to the number of matchings with sub-optimal arms by induction, thus making the regret reach the lower bound related to time horizon $T$ and reward gap $\Delta$ (Appendix D) in matching problem with bandit algorithm Sankararaman et al. (2021).

In a summary, there might be other possible ways to extend the $\alpha$-condition but we present a successful trial to not only give a good extension with similar inclusion relationships but also guarantee a good regret bound.

# 6 EXPERIMENTS

In this section, we verify the experimental results of our MO-UCB-D4 algorithm (Algorithm 1) for decentralized many-to-one matching markets. For all experiments, the rankings of all agents and arms are sampled uniformly. We set the reward value towards the least preferred arm to be $1/N$ and the most preferred one as 1 for each agent, then the reward gap between any adjacently ranked arms is $\Delta = 1/N$. The reward for agent $j$ matches with arm $k$ at time $t$ $X_{j,k}(t)$ is sampled from $\text{Ber}(\mu_{j,k})$. The capacity is equally set as $q = N/K$. We investigate how the cumulative regret and cumulative market unstability depend on the size of the market and the number of arms under three different unique stability conditions: *Serial Dictatorship*, *SPC*, $\tilde{\alpha}$-*condition*. The former cumulative regret is the total mean reward gap between the stable matching result and the simulated result, and the latter cumulative unstability is defined as the number of unstable matchings in round $t$. In our experiments, all results are averaged over 10 independent runs, hence the error bars are calculated as standard deviations divided by $\sqrt{10}$.

**Varying the market size.** To test effects on cumulative regret and cumulative unstability, we first vary $N$ with fixed $K$ with market size of $N \in \{10, 20, 30, 40\}$ agents and $K = 5$ arms. The number of rounds is set to be $100,000$. The cumulative regrets in Figure 2(a)(c)(e) show an increasing trend with convergence as the number of agents increases under these three conditions. When the number of agents increases, there is a high probability of collisions among agents, resulting in an increase of

cumulative regret. Similar results for cumulative unstability are shown in Figure 2(b)(d)(f). When $N$ is larger, the number of unstable pairs becomes more. With the increase of the number of rounds, both two indicators increase first and then tend to be stable. The jumping points are caused by multi-phases setting of MO-UCB-D4 algorithm.

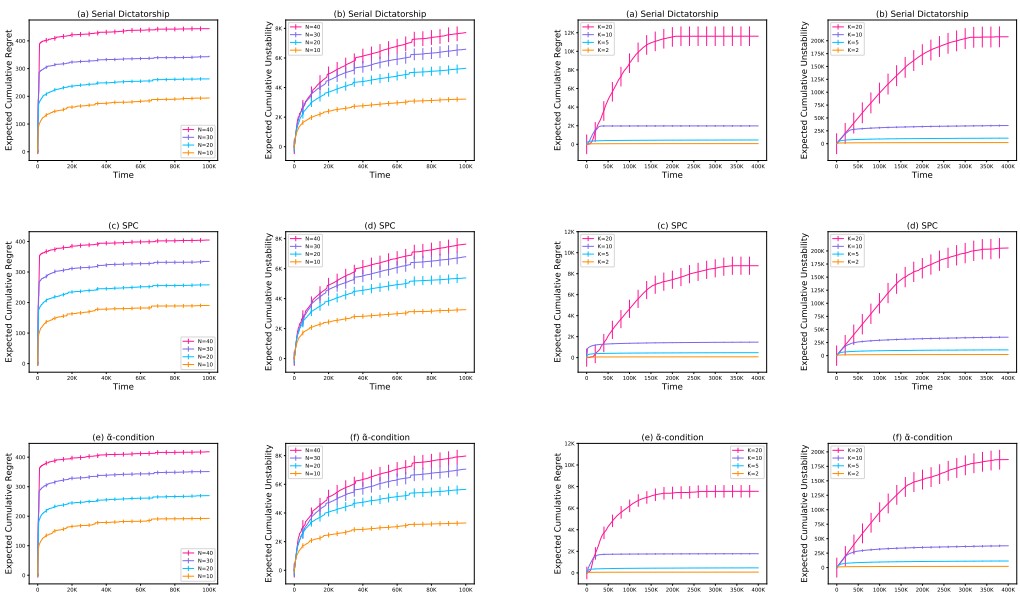

Figure 2: Cumulative regret and cumulative unstability of MO-UCB-D4 of size with $N \in \{10, 20, 30, 40\}$ and the number of arms $K = 5$ under *Serial Dictatorship*, *SPC*, $\tilde{\alpha}$-*condition*.

Figure 3: Cumulative regret and cumulative unstability of MO-UCB-D4 of size with $K \in \{2, 5, 10, 20\}$ under *Serial Dictatorship*, *SPC*, $\tilde{\alpha}$-*condition*.

**Varying arm capacity.** The number of arms $K$ is chosen by $K \in \{2, 5, 10, 20\}$, with $N = 20$ and $q = N/K$. The number of rounds we set is $400,000$. With the increase of $K$, both the cumulative regret in Figure 3(a)(c)(e) and the cumulative unstability in Figure 3(b)(d)(f) increase monotonously. When $K$ increases, the capacity $q_k$ for each arm $k$ decreases, and then the number of collisions will increase, which leads to an increase of cumulative regret. And it also leads to more unstable pairs, which needs more communication blocks to converge to a stable matching. Under these three conditions, the performances of the algorithm are similar.

## 7 CONCLUSIONS

We are the first to study the bandit algorithm for the many-to-one matching market under the unique stable matching. This work focuses on a decentralized market. A new $\tilde{\alpha}$-*condition* is proposed to guarantee a unique stable outcome in many-to-one market, which is more general than existing uniqueness conditions like *SPC*, *Serial Dictatorship* and could recover the usual $\alpha$-*condition* in one-to-one setting. We propose a phase-based algorithm of MO-UCB-D4 with arm-elimination, which obtains $O\left(\frac{NK \log(T)}{\Delta^2}\right)$ stable regret under $\tilde{\alpha}$-*condition*. By carefully defining a mapping from arms to the least preferred agent in its stable matched set, we could effectively correspond arms and agents by individual-to-individual. A series of experiments under two environments of varying the market size and varying arm capacity are conducted. The results show that our algorithm performs well under *Serial Dictatorship*, *SPC* and $\tilde{\alpha}$-*condition* respectively.

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

## A    RELATED WORKS

The study of matching markets has a long history in economics and operation research Bogomolnaia & Moulin (2001); Bade (2020); Roth & Sotomayor (1992) with real applications like school enrollment, labor employment, hospital resource allocation, and so on Abizada (2016); Ma (2010); Roth (1986); Hatfield et al. (2014). A salient feature of market matching is making decisions for competing players on both sides Thompson (1933); Gale & Shapley (1962). MAB is an important tool to study matching problems under uncertainty to obtain a maximum reward, and upper confidence bound algorithm (UCB) Auer et al. (2002) is a typical algorithm, which sets a confidence interval to represent uncertainty.

This paper contributes mainly to intersection of MABs and two-sided matching markets literature. We analyze recent works in this direction. After Das & Kamenica (2005) proposed to apply MAB in learning preference, the learning uncertain matching system provided inspiration for the design of online platform, and then there was a series of algorithm design Liu et al. (2020a;b); Sankararaman et al. (2021); Basu et al. (2021); Gunn et al. (2022); Malgonde et al. (2020); Johari et al. (2021). In a general centralized market without conflicts, Liu et al. (2020a) applied the common ETC and UCB algorithms to the matching market, and obtained the regret order of $O(\frac{NK\log(T)}{\Delta})$. Following this, a more general market, decentralized one, was studied by traditional UCB algorithm, and obtain a $O(\frac{\exp(N^4)N^5K^2\log^2(T)}{\Delta})$ regret by setting a delay parameter to reduce collisions among agents. By limiting preferences, we can get algorithms that have better convergence or can learn information about unknown preferences. Under Serial Dictatorship condition, Sankararaman et al. (2021) proposed an phased UCB algorithm with global communication to solve decentralized market with nonlocal information. As Serial Dictatorship condition is too strong, a weaker Uniqueness Consistency condition is applied in this online data-driven market Basu et al. (2021). Under the conditions on preferences, the regret bound in decentralized matching is reduced to $O(\frac{NK\log(T)}{\Delta})$. However, these valuable articles focused on one-to-one matching that one arm can accept only one agent as his stable pair. Motivated by these, we extend works not only to a many-to-one setting, but also under a weaker uniqueness condition which is first introduced by this work.

In terms of uniqueness conditions, a flurry of works proposed some descriptive conditions in one-to-one setting, like the *Serial Dictatorship* Sankararaman et al. (2021), the *No Crossing Condition (NCC)* Clark (2006), the *Sequential Preference Condition (SPC)* Eeckhout (2000), the $\alpha$-*Condition* Karpov (2019). However, a few of works concentrated on the unique stable property in many-to-one market. Some exiting conditions are *SPC*, Reny (2021), *Aligned Preference*, *Serial Dictatorship Top-top match* and *Acyclicity* Niederle & Yariv (2009); Akahoshi (2014); Reny (2021), which are strong that are not universal in algorithm design.

The research on many-to-one market is a relatively meaningful work recently. Leaning preferences and form a stable matching are also key features in this setting Jagadeesan et al. (2021). Altinok (2019); Özkan & Ward (2020); Johari et al. (2021) studied dynamic many-to-one matching. For one thing, their concerns provide motivation for our work, for another, they also provide more latent future directions for the application of MAB in matching.

## B    ANALYSIS FOR OUR $\tilde{\alpha}$-CONDITION

### B.1    MAPPING UNDER $\tilde{\alpha}$-CONDITION

To connect two sides of the market, we define a mapping $lr$ as $lr(i) = \max\{j : A_j \in \gamma^*(m^*(i)), j \in [N]\}$, from agent index in the left order to agent index in the right order under $\tilde{\alpha}$-*condition* since arms in the right order can select more than one agents. From Theorem 1, the stable matching for arm $k$ is its first $q_k$ preferred agents $\gamma^*(c_k) = \mathcal{A}_k$. Recall that the preference is strict. Denote that the first $q_k$ agents are ranked as $\mathcal{A}_k^{(1)} \succ \mathcal{A}_k^{(2)} \succ \cdots \mathcal{A}_k^{(q_k)}$. Then the rule of the mapping $lr$ in the right order we set is as follows: the mapping for arm $k \in [K]$ is the least preferred one among its most preferred $q_k$ agents, that is, $A_{lr(k)} = \mathcal{A}_k^{(q_k)}$. And the intuitive representation can be seen in Figure 4. If we assume that $c_{i_2} = c_1$, then the right order can be seen form the figure and $lr(q_1 + 1) = \cdots = lr(q_1 + q_{c_1}) = q_{c_1}$ holds.

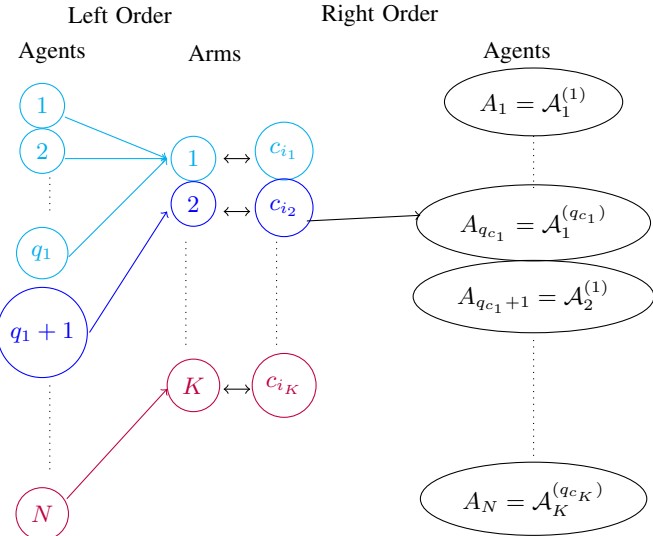

Figure 4: The mapping from the left order to the right order (assume that $c_{i_2} = c_1$)

## B.2 PROOF FOR REGRET ANALYSIS UNDER $\tilde{\alpha}$ - CONDITION

The proof idea is mainly as follows. We construct phases with good properties and denote that the time point of agent $j$ reaching its *good phase* by $F_{\alpha j}$. From phase $F_{\alpha j}$ on-wards, agent $j+1$ will find the globally dominated arm set $G^*_{j+1}$ and will eliminate arm $m^*(j)$ according to Algorithm 1. Then the process of each agent is divided into two stages: before $F_{\alpha j}$ and after $F_{\alpha j}$. After $F_{\alpha j}$, according to the causes of regret, it is divided into four blocks: collision, local deletion, communication, and sub-optimal play. Phases before $F_{\alpha j}$ can be bounded by induction.

We first give some notations and definitions:

**Rank for Each Agent** Recall that if arm $k$ prefers agent $j$ over $j'$, we denote $j \succ_k j'$. And under $\tilde{\alpha}$-condition, the stable matched arm $m^*(j)$ for agent $j$ is agent $j$'s most preferred arm among remaining arms who still have vacant seats within its capacity. Denote the agents that match with the stable matched arm of agent $j$ by $\gamma^*(m^*(j))$.

**Classification of arm sets** *The dominated arms set* $\mathcal{D}_j = \{m^*(j') : j' \succ_{m^*(j')} j\}$ means the stable matched arms of agents who are more preferred by these arms than agent $j$, and *the globally dominated arms set* under stable matching $m^*$ is $G^*_j$, a subset of $\mathcal{D}_j$. Global deletion here follows the left order. Recall that $\mathcal{O}_j[i]$ is the best arm for agent $j$ in phase $i$. In Algorithm 1, the estimated dominated arms set in phase $i$ is $\mathcal{D}_j[i] = \{\mathcal{O}_{j'}[i] : j' >_{\mathcal{O}_{j'}[i]} j\}$ and the globally dominated arms in each phase $i$ $G_j[i] \subset \mathcal{D}_j[i]^2$. For each arm $k \notin G^*_j$, we give the definition of the *blocking agents* for arm $k$ and agent $j$: $\mathcal{B}_{j,k} = \{j' : j' \succ_k j, k \notin G^*_j\}$, which contains agents more preferred by arm $k$ than $j$. The *hidden arms* for agent $j$ is $\mathcal{H}_j = \{k : k \notin G^*_j\} \cap \{k : \mathcal{B}_{j,k} \neq \emptyset\}$.

Under *SPC* condition, the stable matched pair is also the best arm for each agent, and agents that arm $k$ matches with are its $q_k$ most preferred agents. It can be easily understood by the definition of *Top-top match*. While under our $\tilde{\alpha}$-condition, the stable results may not be the best choices for the two sides. We then define a set $NTT(j)$, in which each arm is a stable matched arm for some other agents $\mathcal{A}_{j'}$, is a sub-optimal arm for $j$, and $j$ is preferred by that arm than its stable matched pairs $\gamma^*(k)$. The $NTT(j)$ set can be understood as "not *Top-top match*" stable results, and it can be mathematically expressed as

$$NTT(j) = \left\{ k : k \in [K], \mu_{j,k} < \mu_{j,m^*(j)}, \exists j' \notin \gamma^*(m^*(j)), s.t. k = m^*(\mathcal{A}_{j'}) \text{ and } j \succ_k \gamma^*(k) \right\},$$

---

² We can obtain $\mathcal{D}_j[i] = G_j[i]$ in the one-to-one setting

where $j \succ_k \gamma^*(k)$ means that $k$ prefers $j$ than any agents in $\gamma^*(k)$.

**Phases with Good Properties** In the decentralized market with limited information, estimating preferences of other agents is challenging, thus we set a communication block. This block for agent $j$ is mainly to judge the dominated arms of agents that rank higher than $j$, where the dominated arm is measured as the arm with the most number of times matched with each agent. Under our $\tilde{\alpha}$-condition, the most preferred arm is not necessarily the stable matched result, hence if arms in $NTT(j)$ match too many times with $j$, agents cannot distinguish the preference of agent $j$. During the time period with limitation of arms in the $NTT(j)$, other agents can identify the preferences of $j$, which helps to reduce conflicts.

**Definition 3.** *We say phase $i$ is a **Warm-up Phase** for some $j \in [N]$ under $\tilde{\alpha}$-condition if the following conditions hold for each arm $k \in NTT(j)$:*

*(i) arm $k$ is matched with agent $j$ at most $\frac{10\alpha i}{\Delta_{j,k}^2}$ in phase $i$, where $\alpha$ is a parameter of UCB index (line 7 in Algorithm 1);*

*(ii) arm $k$ is not agent $j$'s most matched arm in phase $i$.*

According to it, we introduce the *Unlocked phase ($U_j$)* that all phases on and after it, agents $A_1$ to $A_j$ are all into warm-up phase. Let $i_1 = \min\left\{i : (N-1)\frac{10\alpha i}{\Delta^2} < \theta 2^{(i-1)}\right\}$, where $\Delta$ is the minimum reward gap, and

$$\mathbb{1}_W[i,j] = \begin{cases} 1, & \text{phase } i \text{ is a warm-up phase for agent } j; \\ 0, & \text{otherwise.} \end{cases}$$

$$U_j = \max\left(i_1, \min\left(\left\{i : \prod_{j'=1}^{lr(j)-1} \prod_{i' \geq i} \mathbb{1}_W[i', A_j'] = 1\right\} \cup \{\infty\}\right)\right).$$

**Definition 4.** *We say phase $i$ is a $\tilde{\alpha}$-**Good Phase** for some $j \in [N]$ under $\tilde{\alpha}$-condition if the following are all satisfied:*

*(i) The globally dominated arms for agent $j$ are globally deleted in phase $i$. Then, $G_j[i] = G_j^*$ holds.*

*(ii) The phase $i$ is a warm-up phase for all agents in $\mathcal{L}_j = \{j' : m^*(j) \in NTT(j')\}$.*

*(iii) For each arm $k \notin G_j^* \cup m^*(j)$ (neither be globally deleted nor stable matched arm of agent $j$), arm $k$ is successfully matched with agent $j$ in phase $i$ at most $\frac{10\alpha i}{\Delta_{j,k}^2}$ times.*

*(iv) The stable matched arm $m^*(j)$ is selected the most number of times in phase $i$.*

The definition of $\tilde{\alpha}$-*Good Phase* is naturally to be brought up that during this phase, agent $j$ has collisions with low probability. When agent $j$ selects an arm competing with a more preferred agent by this arm, it receives zero reward with high probability (w.h.p.), thus condition $(i)$ in Definition 4 is necessary for a lower regret. Recall that the stable matched pair may not be the best pair for $j$, $(ii)$ aims to limit arms in other agents' $NTT$ sets to avoid too many conflicts. And $(iii)$, $(iv)$ are beneficial for other agents to estimate the stable matching of agent $j$. Similarly, we define $\tilde{\alpha}$-*Low Collision Phase* as Basu et al. (2021):

**Definition 5.** *We say phase $i$ is a $\tilde{\alpha}$-**Low Collision Phase** for agent $j$ under $\tilde{\alpha}$-condition if:*

*(i) Phase $i$ is a $\tilde{\alpha}$-Good Phase for agent 1 to agent $j$;*

*(ii) Phase $i$ is a $\tilde{\alpha}$-Good Phase for agent $j' \in \cup_{k \in \mathcal{H}_j} \mathcal{B}_{j,k}$.*

Define that

$$F_{\alpha j} = \max\left(i_1, \min(\{i : \prod_{i' \geq i}\left(\prod_{j'=1}^{j-1} \mathbb{1}_{G_\alpha}[i', j']\right)\left(\prod_{j'' \in \mathcal{L}_j} \mathbb{1}_W[i', j'']\right) = 1) \cup \{\infty\}\right), \quad (1)$$

and

$$V_{\alpha j} = \max\left(i_1, \min(\{i : \prod_{i' \geq i} \mathbb{1}_{LC_\alpha}[i', j] = 1\} \cup \{\infty\})\right), \quad (2)$$

where the definitions of $\mathbb{1}_{LC_\alpha}[i,j]$ and $\mathbb{1}_{G_\alpha}[i,j]$ is similar to $\mathbb{1}_W[i,j]$.

Hence, all phases on and after phase $F_{\alpha j}$ are $\tilde{\alpha}$-*Good Phase* and all phases after phase $V_{\alpha j}$ are $\tilde{\alpha}$-*Low Collision Phase* for agent $j$. Hence, $\mathbb{1}_W[i,j]$, $\mathbb{1}_{LC_\alpha}[i,j]$ and $\mathbb{1}_{G_\alpha}[i,j]$ are the indicator to represent whether phase $i$ is a warm-up phase, $\tilde{\alpha}$-low deletion phase and $\tilde{\alpha}$-good phase respectively.

Before we give the complete proof of the regret bound in Theorem 3, we propose some propositions.

**Proposition 1.** *The stable matched arm $m^*(j)$ for agent $j$ can be blocked by agents in $\mathcal{L}_j$, where*
$$\mathcal{L}_j = \left\{ j' : m^*(j) \in NTT(j') \right\}.$$

*Proof.* Assume that we have stable matching $m^*$. By contradiction, if $j \succ_{m^*(j')} j'$ but $\mu_{j,m^*(j)} < \mu_{j,m^*(j')}$, then $(j, m^*(j'))$ forms a blocking pair since they prefer each other than matched one but they are unmatched, this leads to the instability of $m^*$. So, if $j \succ_{m^*(j')} j'$, then $\mu_{j,m^*(j)} > \mu_{j,m^*(j')}$ under the stable matching. Thus, if $j' \succ_{m^*(j)} j$, then $\mu_{j',m^*(j')} > \mu_{j,m^*(j)}$, then $m^*(j) \in NTT(j')$. $\qquad\square$

Proposition 1 tells us that $m^*(j)$ can be blocked only by agents in $\mathcal{L}_j$, and the next proposition gives the range of $\mathcal{L}_j$.

**Proposition 2.** *For each agent $j \in [N]$, $\mathcal{L}_j \subseteq \bigcup_{j'=1}^{lr(j)-1} \mathcal{A}_{j'}$*

*Proof.* Under $\tilde{\alpha}$-condition, for $\forall k < j \leq K$, $c_k \in [K]_r$, $A_j \in [N]_r$, $\gamma^*(c_k) \succ_{c_k} A_j$. And by Theorem 1, $\gamma^*(c_k) = \mathcal{A}_k$. Therefore, for $\forall j, j' \in [N]$, and $j < j'$, $A_j \succ_{m^*(A_j)} A_{j'}$. In particular, for any $j' > lr(j)$, we have $j = A_{lr(j)} \succ_{m^*(j)} A_{j'}$. This implies that for $\forall j' \geq lr(j)$, we can not obtain $j' \succ_{m^*(j)} j$, hence $m^*(j) \notin NTT(j')$, that is, for $\forall j' \geq lr(j)$, $j' \notin \mathcal{L}_j$. Then $\mathcal{L}_j \subseteq \cup_{j'=1}^{lr(j)-1} \mathcal{A}_{j'}$. $\qquad\square$

**Proposition 3.** *For each agent $j \in [N]$, $F_{\alpha_j} \leq \max\left\{ U_{(lr(j)-1)}, \max(F_{\alpha_{j'}} : 1 \leq j' \leq j-1) \right\}$ happens with probability* 1.

*Proof.* By the definition of $U_j$, we know that on and after phase $U_{(lr(j)-1)}$, all agents $\{\mathcal{A}_{j'} : j' = 1, 2, \cdots, lr(j) - 1\}$ are in warm-up phase. By proposition 2, the set of deadlock agents as $\mathcal{L}_j \subseteq \cup_{j'=1}^{lr(j)-1} \mathcal{A}_{j'}$. Hence, all agents in $\mathcal{L}_j$ are also in warm-up phase on and after $U_{lr(j)-1}$. Further, the agents 1 to $(j-1)$ are in $\tilde{\alpha}$-good phase from phase $\max\{F_{\alpha j'} : 1 \leq j' \leq j-1\}$ onwards. Then the proposition holds w.p.1. $\qquad\square$

As the events decomposition for regret minimization block in Lemma 6 requires that $m^*(j)$ always exit and will not be deleted, it is important to find conditions or a certain phase with good properties to guarantee that $m^*(j)$ will not be globally deleted or locally deleted. The next lemma give us theoretical guarantee.

**Lemma 3.** *Let $i_1 = \min\left\{ i : (N-1)\frac{10\alpha i}{\Delta^2} < \theta 2^{i-1} \right\}$, for any phase $i$ ($i \geq i_1$) and any agent $j \in [N]$, the following properties holds.*

    *(a) If phase $i$ and $(i-1)$ are warm-up phases for all $j' \in \mathcal{L}_j$, then $m^*(j)$ will not be globally deleted or locally deleted almost surely, i.e. $m^*(j) \notin \mathcal{L}_j[i] \cup G_j[i]$.*

    *(b) If phase $i \geq \min\left\{ U_{(lr(j)-1)}, F_{\alpha_j} \right\} + 1$, then $m^*(j) \notin \mathcal{L}_j[i] \cup G_j[i]$ a.s.*

    *(c) If phase $i \geq V_{\alpha_j} + 1$ is a low collision phase for agent $j$ then $\mathcal{L}_j[i] = \emptyset$ a.s.*

*Proof.* (i) All agents $j'$ can block arm $m^*(j)$ are in $\mathcal{L}_j$ by Proposition 1. And $m^*(j) \in NTT(j')$ for any agent $j' \in \mathcal{L}_j$ due to the definition of $\mathcal{L}_j$. Therefore, if all agents in $\mathcal{L}_j$ are in warm-up phase in phase $(i-1)$, then $m^*(j) \notin G_j[i]$ because by the definition of warm-up phase for agent $j'$ and

$m^*(j) \in NTT(j')$, so $m^*(j)$ is not agent $j'$ 's most matched arm. Hence, $m^*(j) \notin G_j[i]$. furthermore, the total number of times the arm $m^*(j)$ can be deleted is at most $\left(\sum_{i=1}^{lr(j)-1} q_i - 1\right) \frac{10\alpha i}{\Delta_{j,k}^2}$ for any $i \geq i_1$, which is less than the local deletion threshold. So $m^*(j) \notin L_j[i] \cup G_j[i]$ after phase $i_1$.

(ii) (a) $\mathcal{L}_j \subseteq \cup_{j'=1}^{lr(j)-1} \mathcal{A}_{j'}$ holds by Proposition 3, this implies that for phase $i \geq U_{lr(j)-1} + 1$ (i.e. $i - 1 \geq U_{lr(j)-1} + 1$) is a warm-up phase for all agents in $\mathcal{L}_j = \{j' : m^*(j) \in NTT(j)\}$.

(b) By the definition of $F_{\alpha j}$, all agents in $\mathcal{L}_j = \{j' : m^*(j) \in NTT(j)\}$ are in warm-up phase for phase $i \geq F_{\alpha j+1}$.

By (a), (b) and (i) we know that (ii) holds.

(iii) It can easily check by the definition of $V_{\alpha j}$. □

### B.3 PROOF FOR THEOREM 3

After defining $F_{\alpha j}$ and $V_{\alpha j}$[3], we divide the whole process into two main modules: the process before phase $F_{\alpha j}$ and after $F_{\alpha j}$. We denote $S_i$ by the beginning time point of phase $i$. The regret during time period $[S_{F_{\alpha j}}, T]$ can be decomposed by four blocks: Local Deletion Block, Communication Block, Collision Block and Sub-optimal Block. The regret during time period $[0, S_{F_{\alpha j}}]$ can be bounded by induction with $j$ (Lemma 7).

**Local Deletion Block.** Lemma 3 implies that there is no collision after phase $V_{\alpha j}$, so we only need to consider the regret from $F_{\alpha j} + 1$ to $V_{\alpha j}$. Following our algorithm, there is at most $\theta 2^{i-1}$ collisions when pulling an arm from the set $\mathcal{H}_j$ in each round. This amounts to

$$\sum_{i=(F_{\alpha j}+1)}^{V_{\alpha j}} \sum_{k \in \mathcal{H}_j} \theta \cdot 2^{i-1} \leq \sum_{i=(F_{\alpha j}+1)}^{V_{\alpha j}} \theta|\mathcal{H}_j| \cdot 2^{i-1}$$
$$< \frac{1 - 2^{V_{\alpha j}-1}}{1-2}\theta|\mathcal{H}_j| = (2^{V_{\alpha j}-1} - 1)\theta|\mathcal{H}_j|$$
$$= S_{V_{\alpha j}} \cdot \theta|\mathcal{H}_j| \leq \min(S_{V_{\alpha j}}, 1) \cdot \theta|\mathcal{H}_j|.$$

**Communication Block.** In the communication block, there are $N$ sub-blocks, and the duration of each sub-block is $K$. Agent $j$ pulls arm 1, arm 2, $\cdots$, arm $K$ in order in the $j$-th block and pulls $\mathcal{O}_j[i]$ in other blocks, where $\mathcal{O}_j[i]$ is the arm that it matched the most times in the regret minimization block in phase $i$. The best arm for agent $j$ is not played in all but $(K-1)$ number of steps for each communication phase after phase $F_{\alpha j} + 1$, and other agents $j'$ collide at most once after phase $V_{\alpha j}$ (since each of them enters good phase). Hence, the regret comes from communication block is $\left((K - 1 + |\mathcal{B}_{j,m^*(j)}|) \log_2(T) + NK\mathbb{E}\left[V_{\alpha j}\right]\right)$.

**Collision Block.** The regret caused by collision from phase $F_{\alpha j} + 1$ to $V_{\alpha j}$ has been included in the previous communication block (the regret of the period during $F_{\alpha j} + 1$ and $V_{\alpha j}$ is relatively loose), so we only consider the regret after phase $V_{\alpha j}$. After phase $V_{\alpha j} + 1$, regret comes from the collision between agent $j$ and the agents in the set $\mathcal{B}_{j,k}$. And by the definition of $V_{\alpha j}$, agent $j$ and agent $j' \in \mathcal{B}_{jk}$ have deleted dominated arms for themselves, this leads to $\sum_{k \notin G_j^*} \sum_{j' \in \mathcal{B}_{j,k}: k \notin G_{j'}^*} \mu_{j,m^*(j)}\left(N_{j',k}(T) - N_{j',k}(S_{V_{\alpha j}})\right)$. And by lemma 6, the number of the matchings with suboptimal arms can be bounded, and the main resource of regret is bounded as a scale of $O(\frac{NK \log(T)}{\Delta^2})$[4].

---

[3]Under $\tilde{\alpha}$-condition it is no longer the case as agent 1 is not the most preferred agent for arm 1. For agent $A_1$ and its stable match arm $c_1$, $c_1$ may not be the best arm for agent $A_1$ but for arm $c_1$ we have $A_1$ as its best agent. Therefore, agent $A_1$ will not delete it's stable match pair arm $a_1$, but unless global deletion eliminates better arms it will not converge to this arm.

[4]It is $\tilde{\alpha}$-condition that induces a hierarchy in the matching market, which reduces the regret bound from collision block to the number of matchings with sub-optimal arms by induction.

**Sub-optimal Play Block.** From phase $F_{\alpha j} + 1$ on-wards, regret happens for agent $j$ when agent $j$ selects arm $k \notin G_j^* \cup m^*(j)$ and successfully be matched. This amounts to $\sum_{k \notin G_j^* \cup m^*(j)} \Delta_{jk}(N_{jk}(T) - N_{jk}(S_{F_{\alpha j}}))$ regret, and it can be upper bounded by Lemma 6.

Then we illustrate the relationship among those phases with good properties and indicators. We first show that for phases $i \geq U_{\alpha j - 1} + 1$, the probability that phase $i$ is not a Warm-up phase for agent $A_j$ is low. Let

$$i_1 = \min\{i : (N-1)\frac{10\alpha i}{\Delta^2} < \theta 2^{(i-1)}\} \tag{3}$$

$$i_2 = \min\{i : C(i-1) - 1 \leq 2^{i+1}\}, \tag{4}$$

then we have the following lemma.

**Lemma 4.** *For phase $i \geq i^* = \max(8, i_1, i_2)$, and for $\forall j \in [N]$, $\alpha > 1$, then the following holds:*

$$\mathbb{P}\left((\mathbb{1}_W[i, A_j] = 0) \cap (i \geq U_{j-1} + 1)\right) \leq (K-j)2^{-i(\alpha-1)}\left(1 + \frac{64}{\Delta^2}\right).$$

Similarly, we give the relationship between $F_{\alpha j}$ and $\alpha$-Good phase.

**Lemma 5.** *For any agent $j$ and phase $i \geq i^*$, and for $\alpha > 1$, then*

$$\mathbb{P}\left((\mathbb{1}_{G_\alpha}[i, j] = 0) \cap (i \geq F_{\alpha j} + 1)\right) \leq (K-j)2^{-i(\alpha-1)}\left(1 + \frac{64}{\Delta^2}\right).$$

We only give the proof of Lemma 4, and another one can similarly be verified.

*Proof.*

$$\mathbb{P}\left((\mathbb{1}_W[i, A_j] = 0) \cap (i \geq U_{\alpha j - 1} + 1)\right)$$

$$\underset{(i)}{\leq} \mathbb{P}\left(\cup_{k \in NTT(A_j)}\{(N_{A_j,k}[i] - N_{A_j,k}[i-1]) > \frac{10\alpha i}{\Delta_{A_j,k}^2}\} \cap (i \geq (U_{\alpha j - 1} + 1))\right)$$

$$\underset{(ii)}{\leq} \sum_{k \in NTT(A_j)} \mathbb{P}\left((\cup_{t \in S_i}^{(S_{i+1}-1)} N_{A_j,k}(t) = \frac{10\alpha i}{\Delta_{A_j,k}^2}) \cap (I_t(A_j) = k) \cap (i \geq (U_{\alpha j - 1} + 1))\right)$$

$$\underset{(iii)}{\leq} \sum_{k \in NTT(A_j)} \sum_{t \in S_i}^{(S_{i+1}-1)} \mathbb{P}\left((N_{A_j,k}(t) = \frac{10\alpha i}{\Delta_{A_j,k}^2}) \cap (u_{A_j,k}(t-1) > u_{A_j a_j}(t-1))\right)$$

$$\leq |NTT(A_j)|\, 2^{-i(\alpha-1)}(1 + \frac{64}{\Delta^2})$$

$$\leq (K-j)2^{-i(\alpha-1)}(1 + \frac{64}{\Delta^2}).$$

The inequality $(i)$ is because that if phase $i$ is not a Warm-up phase for agent $A_j$, there exists an arm $k \in NTT(A_j)$, which is played more than $\frac{10\alpha i}{\Delta_{A_j,k}^2}$ times in phase $i$. Next, $(ii)$ holds since the probability of union is less than or equal to the sum of probability. By Lemma 3, $m^*(A_j) \notin G_{A_j}[i] \cup L_{A_j}[i]$. Hence, the inequality $(iii)$ holds since $I_t(A_j) = k$ is equivalent to that the UCB index (line 7 in Algorithm 1) of arm $m^*(j) = a_j$ can not be less than arm $k$. $\square$

We now give the upper bound of $\mathbb{E}\left[N_{jk}(T) - N_{jk}(S_{F_{\alpha j}})\right]$, which is helpful to bound the regret resulting from collision block and sub-optimal block.

**Lemma 6.** *For $\forall j \in [N]$, $k \notin G_j^* \cup m^*(j)$, for $\alpha > 1$,*

$$\mathbb{E}\left[N_{j,k}(T) - N_{j,k}(S_{F_{\alpha j}})\right] \leq \phi(\alpha)\frac{8}{\Delta_{j,k}^2} + 1 + \frac{8}{\Delta_{j,k}^2}\left(\alpha \log(T) + \sqrt{\pi \alpha \log(T)} + 1\right).$$

*Proof.* Due to Lemma 3, $m^*(j)$ will not be globally deleted or locally deleted after phase $i \geq (F_{\alpha j} + 1)$. Denote $I_j(t)$ as the arm that agent $j$ pulls at time $t$. After phase $F_{\alpha j}$, the reason for agent $j$ pulling arm $k$ rather than $m^*(j)$ are as follows: (1) the *UCB* index of the optimal arm $m^*(j)$ is less than $\mu_{j,m^*(j)} - \epsilon$; (2) $I_t(j) = k$ and its *UCB* index is larger than $\mu_{j,m^*(j)} - \epsilon$. For any $k \notin G_j^* \cup m^*(j)$ and $\epsilon > 0$,

$$N_{j,k}(T) - N_{j,k}(S_{F_{\alpha j}}) = \sum_{t=S_{F_{\alpha j}}+1}^{T} \mathbb{1}\{I_t(j) = k\}$$

$$\leq \sum_{t=S_{F_{\alpha j}}+1}^{T} \left[ \underbrace{\mathbb{1}\{(u_{j,k}(t) \geq \mu_{j,m^*(j)} - \epsilon) \wedge (I_t(j) = k)\}}_{(a)} + \underbrace{\mathbb{1}\{u_{j,m^*(j)} \leq \mu_{j,m^*(j)} - \epsilon\}}_{(b)} \right].$$

First, we bound $(a)$.

$$\mathbb{E}\left[ \sum_{t=S_{F_{\alpha j}}+1}^{T} \mathbb{1}\left\{ (u_{j,k}(t) \geq \mu_{j,m^*(j)} - \epsilon) \wedge (I_t(j) = k) \right\} \right]$$

$$\leq \mathbb{E}\left[ \sum_{t=S_{F_{\alpha j}}+1}^{T} \mathbb{1}\left\{ \left(\hat{\mu}_{j,k}(t-1) + \sqrt{\frac{2\alpha \log(t)}{N_{j,k}(t-1)}} \geq \mu_{j,m^*(j)} - \epsilon\right) \wedge (I_t(j) = k) \right\} \right]$$

$$\leq \mathbb{E}\left[ \sum_{t=1}^{T} \mathbb{1}\left\{ \left(\hat{\mu}_{j,k}(t-1)\sqrt{\frac{2\alpha \log(T)}{N_{j,k}(t-1)}} \geq \mu_{j,m^*(j)} - \epsilon\right) \wedge (I_t(j) = k) \right\} \right]$$

$$\leq \mathbb{E}\left[ \sum_{s=1}^{T} \mathbb{1}\left\{ \left(\hat{\mu}_{j,k}(s) + \sqrt{\frac{2\alpha \log(T)}{s}} \geq \mu_{j,k} + \Delta_{j,k} - \epsilon\right) \right\} \right]$$

$$\leq 1 + \frac{2}{(\Delta_{j,k} - \epsilon)^2} \left( \alpha \log(T) + \sqrt{\alpha \pi \log(T)} + 1 \right).$$

Then we turn to bound $(b)$

$$\mathbb{E}\left[ \sum_{t=S_{F_{\alpha j}}+1}^{T} u_{j,m^*(j)} \leq \mu_{j,m^*(j)} - \epsilon \right]$$

$$\leq \mathbb{E}\left[ \sum_{t=1}^{T} u_{j,m^*(j)} \leq \mu_{j,m^*(j)} - \epsilon \right]$$

$$\leq \mathbb{E}\left[ \sum_{t=1}^{T} \sum_{s=1}^{T} \mathbb{P}\left( \hat{\mu}_{j,k}(t-1) + \sqrt{\frac{2\alpha \log(t)}{N_{j,k}(t-1)}} \leq \mu_{j,m^*(j)} - \epsilon \right) \right]$$

$$\leq \sum_{t=1}^{T} \sum_{s=1}^{T} \exp\left( -\frac{s}{2}(\sqrt{\frac{2\alpha \log(t)}{s}} + \epsilon)^2 \right)$$

$$\leq \sum_{t=1}^{T} t^{-\alpha} \sum_{s=1}^{T} \exp(-\frac{s\epsilon^2}{2})$$

$$\leq \psi(\alpha)\frac{2}{\epsilon^2}.$$

By choosing $\epsilon = \frac{\Delta_{j,k}}{2}$, we have

$$\mathbb{E}\left[ N_{j,k}(T) - N_{j,k}(S_{F_{\alpha j}}) \right] \leq \psi(\alpha)\frac{8}{\Delta_{j,k}^2} + 1 + \frac{8}{\Delta_{j,k}^2} \left( \alpha \log(T) + \sqrt{\alpha \pi \log(T)} + 1 \right).$$

$$\square$$

We define $lr_{\max}(j) = \max\{lr(j') : 1 \le j' \le j\}$, and $\tilde{F}_j = \max\left(U_{lr_{\max}(j)-1}, \max(\tilde{F}_{j'} : 1 \le j' \le (j-1))\right)$, and $\tilde{F}_j > F_{\alpha j}$. Then we introduce a lemma to bound the probability that a phase $i$ is not an $\tilde{\alpha}$-Good phase when $i \ge F_{\alpha j} + 1$.

**Lemma 7.** *For any $j \in [N]$ and $m \ge 1$, the following hold with $i^*$ ($i^* = \max\{8, i_1, i_2\}$)*

$$\mathbb{E}\left[\tilde{F}_j^m\right] \le 2i_1 + (lr_{\max}(j) + j - 2)\left((i^*)^m + K(1 + \frac{64}{\Delta^2})\right)\frac{2^{-(\alpha-1)(i^*-2)}}{(2^{(\alpha-1)} - 1)^2},$$

$$\mathbb{E}\left[2^{\tilde{F}_j}\right] \le 2i_1 + (lr_{\max}(j) + j - 2)\left(2^{i^*} + K(1 + \frac{64}{\Delta^2})\right)\frac{2^{-(\alpha-1)(i^*-2)}}{(2^{(\alpha-1)} - 1)^2}.$$

The proof is the same as Basu et al. (2021).

Hence, the upper bound of $\mathbb{E}\left[S_{F_{\alpha j}}\right]$ is

$$\mathbb{E}\left[S_{F_{\alpha j}}\right] = \mathbb{E}\left[C(F_{\alpha j} - 1) + 2^{F_{\alpha j}}\right] \le \mathbb{E}\left[C(\tilde{F}_j - 1) + 2^{\tilde{F}_j}\right]$$

$$\le C(2i_1 - 1) + C\left(lr_{\max}(j) + j - 2\right)i^* + \left(lr_{\max}(j) + j - 2\right)2^{i^*}$$

$$+ \left(C + 1\right)\left(lr_{\max}(j) + j - 2\right)K\left(1 + \frac{64}{\Delta}\right)\frac{2^{-(\alpha-1)(i^*-2)}}{(2^{(\alpha-1)} - 1)^2},$$

where $C$ is a constant term.

Then for formula with term $\mathbb{E}\left[S_{V_{\alpha j}}\right]$, we can transform its upper bound to another term related to $\mathbb{E}\left[S_{\tilde{F}_{J_{\max(j)}}}\right]$ since

$$V_{\alpha j} = \max\left(F_{\alpha(j+1)}, \cup_{k \in \mathcal{H}_j \cup_{j' \in \mathcal{B}_{jk}} F_{\alpha j}}\right) \le \max\left(\tilde{F}_{(j+1)}, \cup_{k \in \mathcal{H}_j \cup_{j' \in \mathcal{B}_{jk}} \tilde{F}_{(j+1)}}\right) = \tilde{F}_{J_{\max(j)}}.$$

Hence, $\mathbb{E}\left[S_{V_{\alpha j}}\right] \le \mathbb{E}\left[S_{\tilde{F}_{J_{\max(j)}}}\right]$.

Lastly, the regret can be bounded by the decomposition of $\mathbb{E}\left[S_{F_{\alpha j}}\right]$ and phases after $S_{F_{\alpha j}}$ with properties above, where phases on and after $S_{F_{\alpha j}}$ contain local deletion, collision, communication, sub-optimal play blocks.

$$\mathbb{E}[R_j(T)] \le \mathbb{E}\left[S_{F_{\alpha j}}\right] + \min(\theta|\mathcal{H}_j|, 1)\mathbb{E}\left[S_{V_{\alpha j}}\right] + \left((K - 1 + |\mathcal{B}_{j,m^*(j)}|)\log_2(T) + NK\mathbb{E}\left[V_{\alpha j}\right]\right)$$

$$+ \sum_{k \notin G_j^* } \sum_{j' \in \mathcal{B}_{j,k}: k \notin G_{j'}^*} \frac{8\alpha\mu_{kj^*}}{\Delta_{j'k}^2}\left(\log(T) + \sqrt{\frac{\pi}{\alpha}\log(T)}\right) + \sum_{k \notin G_j^* \cup m^*(j)} \frac{8\alpha}{\Delta_{j,k}}(\log(T) + \sqrt{\frac{\pi}{\alpha}\log(T)})$$

$$+ NK\left(1 + (\phi(\alpha) + 1)\frac{8\alpha}{\Delta^2}\right) \le \sum_{k \notin G_j^*} \sum_{j' \in \mathcal{B}_{j,k}: k \notin G_{j'}^*} \frac{8\alpha\mu_{kj^*}}{\Delta_{j',k}^2}\left(\log(T) + \sqrt{\frac{\pi}{\alpha}\log(T)}\right)$$

$$+ \sum_{k \notin G_j^* \cup m^*(j)} \frac{8\alpha}{\Delta_{j,k}}\left(\log(T) + \sqrt{\frac{\pi}{\alpha}\log(T)}\right) + c_j \log_2(T)$$

$$+ O\left(\frac{N^2K^2}{\Delta_{\min}^2} + \left(\min(1, \theta|\mathcal{H}_j|)f_{\tilde{\alpha}}(J_{\max}(j)) + f_{\tilde{\alpha}}(j) - 1\right)2^{i^*} + N^2Ki^*\right).$$

## C  PROOF FOR UNIQUE STABLE CONDITIONS

### C.1  UNIQUENESS CONDITIONS IN ONE-TO-ONE MATCHING.

There are many existing conditions that guarantee the unique stable matching in one-to-one setting, like the *Serial Dictatorship* Sankararaman et al. (2021), the *No Crossing Condition (NCC)* Clark

(2006), the *Sequential Preference Condition (SPC)* Eeckhout (2000), the $\alpha$-*Condition* Karpov (2019). Previous works tell us that *top-top match* and *SPC* condition can lead to a unique stable matching in both one-to-one Niederle & Yariv (2009); Clark (2006) and many-to-one setting Reny (2021). Niederle & Yariv (2009) use the *Top-top match* property instead of $\alpha$-*reducibility* [5] for the same meaning in the one-to-one setting. *Serial Dictatorship* in one-to-one setting means that for each agent, the arms are ranked heterogeneously, in an increasing order of arm-means which is different for each agent-arm pair while the agents are ranked homogeneously across all arms, and vice versa. Followed by Romero-Medina & Triossi (2013); Niederle & Yariv (2009), we know that *Aligned preference* is equal to *Serial dictatorship* in marriage problem as they are both equivalent to no cycle property. And *NCC* and *Serial Dictatorship* are not mutually inclusive, which can be seen in Clark (2006). Hence, the relationship can be represented intuitively in figure 5:

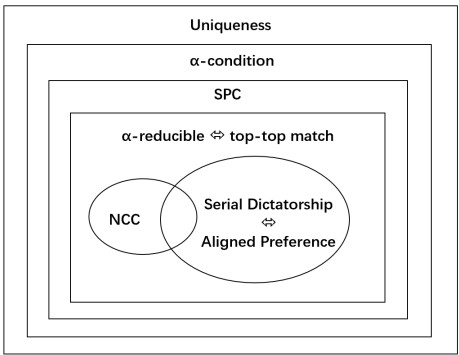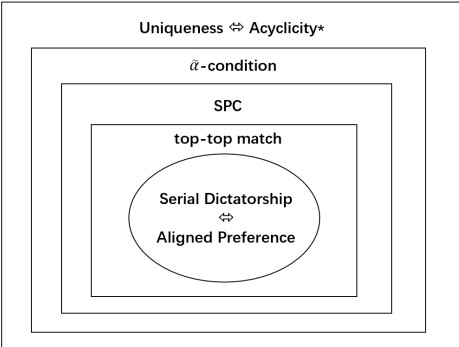

Figure 5: Relations of Unique Stable Conditions in One-to-one (left) and Many-to-one (right) Setting.

## C.2 Uniqueness Conditions in Many-to-one Setting.

In this section, we focus on conditions that guarantee the unique stable matching in the many-to-one setting, such as *SPC*, Reny (2021), *Aligned Preference*, *Serial Dictatorship Top-top match* and *Acyclicity* Niederle & Yariv (2009); Akahoshi (2014); Reny (2021) and give the proof of the relationships among uniqueness conditions[6].

**Definition 6.** *(Aligned Preference.) In a many-to-one market* $\mathcal{M} = (\mathcal{K}, \mathcal{J}, \mathcal{P})$, $\mathcal{K} = (k)_{k \in [K]}$, $\mathcal{J} = (j)_{j \in [N]}$, *if the preference profile* $\mathcal{P}$ *satisfies*

$$\forall k \in \mathcal{K}, j \succ_k j', \forall j < j' \quad (1.a)$$
$$\forall j \in \mathcal{N}, k \succ_j k', \forall k < k' \quad (1.b)$$

*then the market has aligned preference. The one-to-one setting has the same definition.*

**Definition 7.** *(Serial Dictatorship) We say that if all arms (school) have the same preference for agents (students), while agents' preferences are heterogeneous (vice versa), then the system satisfies serial dictatorship.*

**Definition 8.** *(Top-top Match) A stable pair* $(k, j)$ *is a Top-top match for sub-market* $\mathcal{M}' \in \mathcal{M}$ *if, for arm* $k$, *agent* $j$ *is the favorite candidate in* $\mathcal{M}'$, *and vice versa.*

**Definition 9.** *(SPC) SPC condition in the many-to-one setting Reny (2021) is to require the existence of a sequence of agents* $1, 2, \cdots, N$ *in which each agent appears once, and a sequence of arms* $1, 2, \cdots, K$ *in which each arm appears once for each seat in its capacity, such that* $k \succ_j k'$ *for every* $k' > k$ *and* $j \in [N]$; *in addition, such that* $j \succ_k j'$ *for every* $j' > j$ *and* $k \in [K]$.

### C.2.1 Proof for Lemma 1.

*Proof.* $\Rightarrow$):

---

[5]Park (2017); Clark (2006) introduce that a matching problem is $\alpha$-*reducible* if there is a top trading single or pair for every sub-problem.

[6]The remark in Niederle & Yariv (2009) tells us that *Aligned Preference* is stronger than *Top-top match* and *SPC* condition.

Table 1: Preference Profiles

|  | (a) Exm1: Companies |  | (b) Exm1: Workers |
|---|---|---|---|

$$c_1: \quad s_1 > s_2 > s_3 > s_4 > s_5 \qquad\qquad s_1: \quad c_1 > c_2 > c_3$$
$$c_2: \quad s_2 > s_3 > s_4 > s_5 > s_1 \qquad\qquad s_2: \quad c_2 > c_3 > c_1$$
$$c_3: \quad s_3 > s_4 > s_5 > s_1 > s_2 \qquad\qquad s_3: \quad c_3 > c_2 > c_1$$
$$s_4: \quad c_3 > c_1 > c_2$$
$$s_5: \quad c_2 > c_1 > c_3$$

|  | (c) Exm2: Companies |  | (d) Exm2: Workers |
|---|---|---|---|

$$c_1: \quad s_1 > s_2 > s_3 > s_4 > s_5 \qquad\qquad s_1: \quad c_1 > c_3 > c_2$$
$$c_2: \quad s_3 > s_2 > s_1 > s_4 > s_5 \qquad\qquad s_2: \quad c_1 > c_2 > c_3$$
$$c_3: \quad s_1 > s_5 > s_2 > s_4 > s_3 \qquad\qquad s_3: \quad c_2 > c_1 > c_3$$
$$s_4: \quad c_1 > c_2 > c_3$$
$$s_5: \quad c_3 > c_2 > c_1$$

**Serial Dictatorship $\Rightarrow$ Aligned Preference.** In order to distinguish the symbols of agents and arms, we consider arms set $\{c_k, k = 1, 2, \cdots, K\}$ and agents set $\{s_j : j = 1, 2, \cdots, N\}$. If arms have the same preference for individual agent, then there is no cycle in the preference of the arm, i.e. there is no case that

$$\exists T, s_0 \succ_{c_0} s_T \succ_{c_T} s_{T-1} \cdots s_1 \succ_{c_1} s_0$$

for $s_0, s_1, \cdots, s_T$ and $c_0, c_1, \cdots, c_T$. Otherwise, assume that there exists the cycle above, then by the same preference of arms, we know that $\succ_{c_0} = \succ_{c_1}$. And then $s_0 \succ_{c_0} s_1$ and $s_1 \succ_{c_1} s_0$, hence $s_0 \succ_{c_0} s_1$ and $s_1 \succ_{c_0} s_0$, which yields a contradiction.

Now we prove that no cycle property implies *Aligned preference*. By contradiction, if there exists a $c_l$ such that $s_k \succ_{c_l} s_j$, for $k > j$, then we can construct a cycle:

$$s_k \succ_{c_l} s_j \succ_{c_j} s_{j-1} \cdots s_{k-2} \succ_{c_{k-1}} s_{k-1} \succ_{c_k} s_k.$$

$\Leftarrow$):

**Aligned Preference $\Rightarrow$ Serial Dictatorship.** We first illustrate that aligned preference leads to no cycle property. By contradiction, if there is a cycle

$$s_1 \succ_{c_1} s_T \succ_{c_T} s_{T-1} \cdots s_2 \succ_{c_2} s_1$$

for some $s_1, s_2, \cdots, s_T, c_1, c_2, \cdots, c_T$ and $T$. It is obvious that it yields $s_1 \succ_{c_1} s_T, T > 1$, which contradicts the aligned principle. Then, if there is no cycle of length two, which implies that all college have the same preferences because all students are acceptable to every college, which induces the group serial dictatorship property.

$\square$

### C.2.2 PROOF FOR THEOREM 2.

**(i) Proof for the relationship between *SPC* and $\tilde{\alpha}$-*condition***

*SPC* states that after eliminating all Top-top match, there is at least one new Top-top match in the remaining system under the restricted preference profile. Then it satisfies $\tilde{\alpha}$-*condition* naturally. However, examples below tell us that SPC can not imply $\tilde{\alpha}$-*condition*. We give two examples to illustrate this relationship where the order that an agent successfully matches with its stable pair corresponds to the left order and right order.

**Example** Consider a market with three companies and five workers. Assume that the preference profile of companies $c_1, c_2, c_3$ and workers $s_1, s_2, s_3, s_4, s_5$ is as follows and the capacities are $2, 1, 2$ respectively for $c_1, c_2, c_3$.

The preference in Table 1 (1(a))(1(b)) satisfies both *SPC* and $\tilde{\alpha}$-*condition* with valid order $\{(c_2, s_2), (c_3, s_3, s_4), (c_1, s_1, s_5)\}$. While preference in Table 1 (1(c))(1(d)) only satisfies $\tilde{\alpha}$-condition with valid left order $\{(c_1, s_1, s_2), (c_2, s_3), (c_3, s_4, s_5)\}$ and right order

$\{(c_2, s_3), (c_1, s_1, s_2), (c_3, s_4, s_5)\}$, and *SPC* does not hold.

**(ii) Proof for the relationship between Unqc and $\tilde{\alpha}$-*condition***

$\Leftarrow$) : **Sufficiency:** If $\tilde{\alpha}$-*condition* holds, then the agent-proposing Gale-Shapley algorithm and the arm-proposing Gale-Shapley algorithm leads to matching $m$ in all consistent restrictions.

$\Rightarrow$) : **Necessity:** We first prove for $K = 2, N = q_1 + q_2$ case. Assume that there are two arms $c_1, c_2$, each has capacity $q_k(k = 1, 2)$ and the agents set $\mathcal{S} = s_1, s_2, \cdots, s_{q_1+q_2}$. By contradiction, assume that Unqc is satisfied while $\tilde{\alpha}$-condition is not. Then we know that not all matching pairs are Top-top match, so there exists an agent $s_k$, $c_1 \succ_{s_k} c_2$, but $s_k$ is not in the agents set that first $q_1$ preferred by $c_1$. The matched result may have two cases:

$$(\underbrace{\cdots\cdots}_{q_1}, c_1) \, and \, (s_k, \underbrace{\cdots\cdots}_{q_2-1}, c_2) \quad (i) \, ,$$

$$(s_k, \underbrace{\cdots\cdots}_{q_1-1}, c_1) \, and \, (\underbrace{\cdots\cdots}_{q_2}, c_2) \quad (ii) \, .$$

We first consider matching $(ii)$. If $s_k$ matches $c_1$, then there must be an agent in $\mathcal{A}_1$ matches with $c_2$. Let's assume that there is an agent $s_\ell \in \mathcal{A}_1$ that matches with $c_2$. There are two situations to discuss at this time. If $c_1 \succ_{s_\ell} c_2$, then $(ii)$ is an unstable matching, which is recorded as case (A); If $s_\ell$ prefers $c_2$ more than $c_1$, then $(ii)$ is a stable matching and is recorded as event (B).

Apply the above two cases (A), (B) to matching $(i)$. In (A), $c_1$ and $s_\ell$ prefer each other, so there is a Top-top match and then $\tilde{\alpha}$-condition is satisfied, and a conclusion contradictory to the hypothesis is derived. In (B), this case will produce two stable matchings, which contradicts Unqc.

We use induction to prove it. Suppose, that for all $(\hat{N}, \hat{K})$, $\hat{N} \le N, \hat{K} \le K$, $N \ge q_1 + q_2 + \cdots + q_K$ the $\tilde{\alpha}$-condition is a necessary condition for the uniqueness consistency. Then we prove for $(N + 1, q_1 + q_2 + \cdots + q_K)$ (similarly, we would have for $(N, q_1 + q_2 + \cdots + q_K + 1)$ and $q_1 + q_2 + \cdots + q_K \ge N$ ). Assume that the newly added agent is $X$, select an agent from the original $N$ agents and record it as $Y$. Let $k_X^*$ and $k_Y^*$ be the arms rank first for $X$ and $Y$ respectively. By the $K = 2, N = q_1 + q_2$ case proved above, we know that $X$ and $Y$ satisfy $\tilde{\alpha}$-condition, hence either $X$ or $Y$ matches with its first ranked arm. The agent matches with its first ranked arm is denoted by $s_1$, and the remaining $N$ agents are $s_2, \cdots, s_N$. Except $k_{s_1}^*$ and stable matched agents for $k_{s_1}^*$, there are $N$ agents and $K - 1$ arms, and $N \ge q_1 + q_2 + \cdots + q_K - q_{k_{s_1}^*}$. From the inductive hypothesis, we can know that $\tilde{\alpha}$-condition is satisfied.

The relationship between $\tilde{\alpha}$-*condition* and $Acyclicity^*$ is illustrated in Section C.2.4.

### C.2.3 DIFFICULTIES FROM *SPC* TO $\tilde{\alpha}$-CONDITION IN REGRET ANALYSIS

When we use the events decomposition for regret minimization block to prove the bound inequality of the number of times agent $j$ is pulled (Lemma 6), it requires that $m^*(j)$ always exit and will not be deleted. Under *SPC* condition, $m^*(j)$ always exits as the stable matched partner is the most preferred one among the remaining market for the certain agent while $\tilde{\alpha}$-condition cannot guarantee this property. Hence, it is important to find conditions or a certain phase with good properties to guarantee that $m^*(j)$ will not be globally deleted or locally deleted. And we consider $F_{\alpha_j}$ and $V_{\alpha_j}$ in Lemma 3 (in Appendix B.2) to solve this problem. And since the stable matched pair is not top-top match in the remaining system under $\tilde{\alpha}$-condition while the answer is true under *SPC*, we introduce a new mapping (Figure 4) to describe the corresponding relationships of stable pairs. In addition, as shown in Figure 1, $Acyclicity^*$ is the weakest condition to ensure uniqueness up to now, and Bettina Klaus and Flip Klijn Klaus & Klijn (2013) point that acyclicity has a tight connection with consistency. Hence, whether we can further weaken $\tilde{\alpha}$-condition and propose a new algorithm remains to study.

### C.2.4 $Acyclicity^*$ Guarantees A Unique Stable Matching

**Definition 10.** *The preference profile of the arm side $\mathcal{P}_c$ has a cycle with length $\ell$ if there exists integer $\ell \geq 2$, $c_1, c_2, \cdots, c_\ell$ are $\ell$ distinct arms and $s_1, s_2, \cdots, s_\ell$ are $\ell$ distinct agents, subset $T_1, T_2, \cdots, T_\ell \subset \mathcal{S} \backslash \{s_1, \cdots, s_\ell\}$ and for any $i \in \{1, 2, \cdots, \ell\}$, the following two conditions are satisfied.*

*(P) $\{s_{i+1}\} \succ_{c_i} \{s_i\} \succ_{c_i} \phi$, where $s_{l+1} \equiv s_1$, and*

*(Q) $|T_i| = q_{c_i} - 1$ and $T_i \subseteq U_{c_i}(s_i)$, where $U_{c_i}(s_i) = \{s : s \succ_{c_i} s_i\}$.*

*If $\mathcal{P}_c$ has no cycle, it satisfies $Acyclicity^*$.*

Akahoshi (2014) pointed that $Acyclicity^*$ is a necessary and sufficient condition for a unique stable matching in many-to-one matching. They study the problem with responsive preference[7] and unacceptable agents and arms may exist on both sides of the market. Under our setting, both two sides are acceptable, and we will prove that $Acyclicity^*$ is also a necessary and sufficient condition for uniqueness in our problem.

**Theorem 4.** *In our setting, our new $\tilde{\alpha}$-condition is a sufficient condition to $Acyclicity^*$ (Theorem 2 (iii)).*

We first see the example above to explain hoe to check whether the $Acyclicity^*$ is satisfied. As mentioned above, the preference profile in Table 1 (1(a))(1(b)) satisfies both *SPC* and *$\tilde{\alpha}$-condition* with valid order $\{(c_2, s_2), (c_3, s_3, s_4), (c_1, s_1, s_5)\}$. We now check that it also satisfies $Acyclicity^*$.

From preference profile (1(a)), we can find four cycle:

  (i) $s_1 \succ_{c_1} s_2 \succ_{c_2} s_1$;
  (ii) $s_2 \succ_{c_2} s_3 \succ_{c_3} s_2$;
  (iii) $s_3 \succ_{c_2} s_1 \succ_{c_1} s_3$;
  (iv) $s_3 \succ_{c_3} s_1 \succ_{c_1} s_2 \succ_{c_2} s_3$;

Condition $(P)$ in Definition 10 is satisfied, and we then illustrate that condition $(Q)$ is not satisfied, thus $Acyclicity^*$ holds. For cycle (i), $T_1, T_2 \subset \mathcal{S} \backslash \{s_1, s_2\}$, $|T_1| = q_{c_1} - 1 = 1$. However, it violates $T_1 \subset U_{c_1}(s_1) = \emptyset$. Similarly, (ii), (iii), (iv) all imply that $Acyclicity^*$ is satisfied. For cycle (iv), $T_1, T_2, T_3 \subset S \backslash \{s_1, s_2, s_3\}$, $|T_1| = q_{c_1} - 1 = 1$ while $T_1 \subset U_{c_1}(s_1) = \emptyset$. Then, this example also satisfies $Acyclicity^*$.

In fact, we can see from the definitions of these two conditions that $Acyclicity^*$ only limits the preferences of the arm side, while *$\tilde{\alpha}$-condition* limits the preferences of both sides of the market. Intuitively, $Acyclicity^*$ is a more general condition. We now give the theoretical proof.

If $\tilde{\alpha}$-condition holds, then $Acyclicity^*$ also holds. By contradiction, if $Acyclicity^*$ is violated, then there is a *cycle* (Definition 10). For preference sequences that can produce stable matchings, as long as there is a *cycle* or a ring structure, we can always construct at least two stable matchings Romero-Medina & Triossi (2013). For example, for fixed agents set $\mathcal{S} = \{s_1, s_2, \cdots, s_N\}$ and arms set $\mathcal{C} = \{c_1, c_2, \cdots, c_K\}$ with preference profile $\mathcal{P}$ and this matching market has stable matching $m^*$. If there is a *cycle* $s_1 \succ_{c_1} s_2 \succ_{c_2} s_1$, for this stable matching $m^*$ containing $(s_1, c_1)$, $(s_2, c_2)$, when other matching pairs remain unchanged, $(s_2, c_1)$, $(s_1, c_2)$ with other pairs can lead to a new stable matching. Thus the uniqueness is violated, and then $\tilde{\alpha}$-condition is also violated.

Conversely, we consider a counterexample that $Acyclicity^*$ holds while $\tilde{\alpha}$-condition may not hold.

From Table 2, we now explain that a market with arms $c_1, c_2, c_3$, agents $s_1, s_2, s_3, s_4, s_5$, and capacity $q = (2, 1, 2)$ with preference (2(a)) and (2(b)) satisfies $Acyclicity^*$ and can lead to a unique stable matching but does not satisfy $\tilde{\alpha}$-condition. We run GS Algorithm in many-to-one market and obtain stable matching $\{(c1; s_2, s_5), (c_2; s_1), (c_3; s_3, s_4)\}$. And $Acyclicity^*$ is easily verified. After eliminating $(c_3; s_3, s_4)$, only $s_1, s_2, s_5, c_1, c_2$ remain in the system, and then the preference profile is represented as (2(c)) and (2(d)) in Table 2. Apparently, this preference can produce two stable matching. Thus, $\tilde{\alpha}$-condition is violated.

---

[7]The *responsive preference* here means that if only one student in the two matchings is different, the college prefers the matching containing the preferred student.

Table 2: Preference Profiles

| (a) Exm3: Arms | (b) Exm3: Agents |
|---|---|

| | |
|---|---|
| $c_1:$ $s_1 > s_2 > s_5 > s_3 > s_4$ | $s_1:$ $c_2 > c_3 > c_1$ |
| $c_2:$ $s_2 > s_1 > s_4 > s_3 > s_5$ | $s_2:$ $c_1 > c_2 > c_3$ |
| $c_3:$ $s_1 > s_3 > s_2 > s_4 > s_5$ | $s_3:$ $c_3 > c_1 > c_2$ |
| | $s_4:$ $c_1 > c_2 > c_3$ |
| | $s_5:$ $c_1 > c_2 > c_3$ |

| (c) Exm3: Arms | (d) Exm3: Agents |
|---|---|

| | |
|---|---|
| $c_1:$ $s_1 > s_2 > s_5$ | $s_1:$ $c_2 > c_1$ |
| $c_2:$ $s_2 > s_1 > s_5$ | $s_2:$ $c_1 > c_2$ |
| | $s_5:$ $c_1 > c_2$ |

**Theorem 5.** *Suppose that $(\mathcal{K}, \mathcal{J}, \mathcal{P})$ are arbitrarily fixed. $\mathcal{P}_c$ and $\mathcal{P}_s$ are the preference profiles of arms and agents respectively. Then, $\mathcal{P}_c$ satisfies Acyclicity\* if and only if there is a unique stable matching in many-to-one setting for each $\mathcal{P}_s$.*

*Proof.* In order to prove this theorem, we first introduce a lemma.

**Lemma 8.** *For a given $\mathcal{P}$, suppose that there are two stable matchings under $\mathcal{P}$: $\mu$, $\mu'$, then Akahoshi (2014)*

- *$|\mu(s)| = |\mu'(s)|$ for each $s \in \mathcal{J}$ and $|\mu(c)| = |\mu'(c)|$ for each $c \in \mathcal{K}$.*

  *Moreover, for each $c \in \mathcal{K}$ with $\mu'(c) \neq \mu(c)$,*

- *$|\mu(c)| = |\mu'(c)| = q_c$;*

- *$\mu(c)\backslash\mu'(c) \neq \emptyset$ and $\mu'(c)\backslash\mu(c) \neq \emptyset$;*

- *if $\mu'(c) \succ_c \mu(c)$, then for each $s' \in \mu'(c)$ and $s \in \mu(c)\backslash\mu'(c)$, $\{s'\} \succ_c \{s\}$.*

$\Rightarrow) :$ **Necessity:** We complete this proof by contradiction. Suppose there are at least two distinct stable matchings under $\mathcal{P}$. From GS algorithm Gale & Shapley (1962), there exists optimal matchings $\mu^s$ and $\mu^c$, s.t. $\mu^c \succ_c \mu^s$ and $\mu^s \succ_s \mu^c$. Under the multi-stability assumption, $\mu^s \neq \mu^c$. Then, $\exists c_0 \in \mathcal{K}$, s.t. $\mu^s(c_0) \neq \mu^c(c_0)$, and by the optimality of $\mu^c$, $\mu^c(c_0) \succ_{c_0} \mu^s(c_0)$. Consider the following algorithm:

- Step 1: Choose $c_1 \in \mathcal{K}$, such that $\mu^s(c_1) \neq \mu^c(c_1)$ and choose $s_2 \in \mathcal{J}$, such that $s_2 \in \mu^c(c_1)\backslash\mu^s(c_1)$. Choose $c_2 \in \mathcal{K}\backslash\{c_1\}$, $\{c_2\} = \mu^s(s_2)$. Go to step 2;

- Step $k$ ($k \geq 2$): Choose $s_{k+1} \in \mathcal{J}$, such that $s_{k+1} \in \mu^c(c_k)\backslash\mu^s(c_k)$ and $c_{k+1} \in \mathcal{K}\backslash\{c_k\}$, s.t. $\{c_{k+1}\} = \mu^s(s_{k+1})$. If $c_{k+1} \in \{c_1, c_2, \cdots, c_k\}$, then the algorithm terminates. If not, go to the next step.

- Result: If the algorithm terminates at Step $\ell$ ($\ell \geq 2$) with $c_{\ell+1} = c_j (j \geq 1)$, then the result is:

  Given the students $\{s_{j+1}, s_{j+2}, \cdots, s_{\ell+1}\}$ and the college $\{c_j, c_{j+1}, \cdots, c_\ell\}$, there is a cycle: $s_{\ell+1} \succ_{c_\ell} s_\ell \cdots\cdots s_{j+2} \succ_{c_{j+1}} s_{j+1} \succ_{c_j} s_j$, then condition $(P)$ is satisfied. Let $T_k = \mu^c(c_k)\backslash\{s_k\}, k \in \{j, j+1, \cdots, \ell\}$, since each agent ultimately matches only one arm, $\mu^c(c_j), \mu^c(c_{j+1}), \cdots, \mu^c(c_\ell)$ are mutually disjoint, then $T_j, T_{j+1}, \cdots, T_\ell$ are disjoint. And by the definition of $T_k, k \in \{j, j+1, \cdots, \ell\}$, $T_k$ does not contain any agent in $\{s_{j+1}, s_{j+2}, \cdots, s_{\ell+1}\}$. By the second property in Lemma 8, $|T_k| = q_{c_k} - 1$ and by the last property, $T_k \subset U_{c_k}(s_k)$.

  Hence, there is a *cycle* (Definition 10), which induces a contradiction.

$\Leftarrow$) : **Sufficiency:** Assume that there exists a cycle $s_{\ell+1} \succ_{c_\ell} s_\ell \cdots s_3 \succ_{c_2} s_2 \succ_{c_1} s_1$, $s_{\ell+1} \equiv s_1$, and $|T_i| = q_{c_i} - 1$, $T_{c_i} \subseteq U_{c_i}(s_i)$, then we construct preference profiles for both arms (Figure C.2.4) and agents (Figure C.2.4):

Table 3: Preference Profile of $\mathcal{K}$.

| note | $c_1$ | $c_2$ | $\cdots\cdots$ | $c_{\ell-1}$ | $c_\ell$ | $c_{\ell+1}$ | $\cdots\cdots$ | $c_k$ |
|------|-------|-------|----------------|--------------|----------|--------------|----------------|-------|
| 1 | $s_2$ | $s_3$ | $\cdots\cdots$ | $s_\ell$ | $s_1$ | $*$ | $\cdots\cdots$ | $*$ |
| 2 | $s_{\ell+2}$ | $s_{\ell+2}$ | $\cdots\cdots$ | $s_{\ell+2}$ | $s_{\ell+2}$ | $*$ | $\cdots\cdots$ | $*$ |
| $\vdots$ | $\vdots$ | $\vdots$ | $\cdots\cdots$ | $\vdots$ | $\vdots$ | $\vdots$ | $\cdots\cdots$ | $\vdots$ |
| | $s_{\ell+1+q_1}$ | $\vdots$ | $\cdots\cdots$ | $s_{\ell+1+q_{\ell-1}}$ | $\vdots$ | $\vdots$ | $\cdots\cdots$ | $\vdots$ |
| $q_i$ | | $s_{\ell+1+q_2}$ | $\cdots\cdots$ | $\vdots$ | $s_{\ell+1+q_\ell}$ | $\vdots$ | $\cdots\cdots$ | $\vdots$ |
| | $s_{\ell+2+q_1}$ | $s_{\ell+2+q_2}$ | $\cdots\cdots$ | $s_{\ell+2+q_{\ell-1}}$ | $1 s_{\ell+2+q_\ell}$ | | $\cdots\cdots$ | |
| | $s_{\ell+3+q_1}$ | $s_{\ell+3+q_2}$ | $\cdots\cdots$ | $s_{\ell+3+q_{\ell-1}}$ | $1 s_{\ell+3+q_\ell}$ | | $\cdots\cdots$ | |
| | $\vdots$ | $\vdots$ | $\cdots\cdots$ | $\vdots$ | $\vdots$ | | $\cdots\cdots$ | |
| | $s_N$ | $s_N$ | $\cdots\cdots$ | $s_N$ | $s_N$ | | $\cdots\cdots$ | |
| The remaining | $s_1$ | $s_1$ | $\cdots\cdots$ | $s_1$ | $s_2$ | | | |
| of $\{s_\ell\}$ | $s_3$ | $s_2$ | $\cdots\cdots$ | $s_2$ | $s_3$ | | | |
| are ranked | $\vdots$ | $s_4$ | $\cdots\cdots$ | $\vdots$ | $\vdots$ | | | |
| at last | $\vdots$ | $\vdots$ | $\cdots\cdots$ | $\vdots$ | $\vdots$ | | | |
| | $s_\ell$ | $s_\ell$ | $\cdots\cdots$ | $s_{\ell-1}$ | $s_\ell$ | | | |

Table 4: Preference Profile of $\mathcal{J}$.

| $s_1$ | $s_2$ | $\cdots\cdots$ | $s_{\ell-1}$ | $s_\ell$ | $s_{\ell+1}$ | $\cdots\cdots$ | $s_N$ |
|-------|-------|----------------|--------------|----------|--------------|----------------|-------|
| $c_1$ | $c_2$ | $\cdots\cdots$ | $s_{\ell-1}$ | $s_1$ | $*$ | $\cdots\cdots$ | $*$ |
| $c_\ell$ | $c_1$ | $\cdots\cdots$ | $c_{\ell-2}$ | $c_{\ell-1}$ | $*$ | $\cdots\cdots$ | $*$ |
| $\vdots$ | $\vdots$ | $\cdots\cdots$ | $\vdots$ | $\vdots$ | $\vdots$ | $\cdots\cdots$ | $\vdots$ |
| $[K]\backslash\{c_1,c_\ell\}$ | $[K]\backslash\{c_2,c_1\}$ | $\cdots\cdots$ | $[K]\backslash\{c_{\ell-1},c_{\ell-2}\}$ | $[K]\backslash\{c_\ell,c_{\ell-1}\}$ | $*$ | $\cdots\cdots$ | $*$ |

Then we can find two distinct matchings $\mu^c$ and $\mu^s$ (Figure C.2.4 and Figure C.2.4), which induce a contradiction.

Table 5: $\mu^c$.

| $c_1$ | $c_2$ | $\cdots\cdots$ | $c_{\ell-1}$ | $c_\ell$ | $c_{\ell+1}$ | $\cdots\cdots$ | $c_K$ |
|-------|-------|----------------|--------------|----------|--------------|----------------|-------|
| $s_2$ | $s_3$ | $\cdots\cdots$ | $s_\ell$ | $s_1$ | $*$ | $\cdots\cdots$ | $*$ |
| $*$ | $*$ | $\cdots\cdots$ | $*$ | $*$ | $*$ | $\cdots\cdots$ | $*$ |

Table 6: $\mu^s$.

| $c_1$ | $c_2$ | $\cdots\cdots$ | $c_{\ell-1}$ | $c_\ell$ | $c_{\ell+1}$ | $\cdots\cdots$ | $c_K$ |
|-------|-------|----------------|--------------|----------|--------------|----------------|-------|
| $s_1$ | $s_2$ | $\cdots\cdots$ | $s_{\ell-1}$ | $s_\ell$ | $*$ | $\cdots\cdots$ | $*$ |
| $*$ | $*$ | $\cdots\cdots$ | $*$ | $*$ | $*$ | $\cdots\cdots$ | $*$ |

$\square$

# D    MORE DISCUSSIONS ABOUT OUR WORK

## D.1    STABILITY IN MANY-TO-ONE SETTING

Stable matchings are always exist in one-to-one market Gale & Shapley (1962) while the answer is not necessarily correct under many-to-one setting Roth & Sotomayor (1992). Roth & Sotomayor (1992) points out that responsive preference (RP) that can refrain from this unexpectation. Our work assume that arm preference profiles are over individuals rather than agents sets, which naturally satisfies RP Sethuraman et al. (2006)[8].

## D.2    SOME DETAILS ABOUT ALGORITHM

**Multi-phases to Reduce Collisions**    In previous work, the CA-UCB algorithm Liu et al. (2020b) was proposed to manage conflicts in the decentralized market combined with the bandit algorithm, but it has limitations for more general preference structures. In CA-UCB, if we set the delay probability for all agents as zero, then agents may fall into infinite loops and cause high regret. To avoid linear regret, the paper of Sankararaman et al. (2021) applies a phased UCB algorithm with arm elimination in the one-to-one setting. Our MO-UCB-D4 algorithm in many-to-one matching is also carried out in multi-phases for conflict management. The multi-phases is to guarantee that the active set in different phases has no inclusion relationship so that if an agent deletes an arm in a phase, this arm can still be selected in the later phases. This ensures when the agent wrongly deletes an arm, it will not lead to linear regret.

**Parameter Selection and Scale**    The parameter $\theta \in (0, 1/K)$ in our MO-UCB-D4 algorithm is chosen for the local deletion threshold. Increasing the threshold leads to higher regret until local deletion vanishes. This happens as more collisions are allowed until an arm is deleted. But a higher threshold allows for quick detection of the stable matched arms. However, decreasing the threshold results in a more aggressive deletion and then lower regret from collision each phase, at a cost of longer detection time for the stable matched arms. Therefore, there is a trade-off when choosing $\theta$ and we can design an algorithm to iteratively update $\theta$ based on the previous information.

**Baseline experimental design**    Although our work mainly focuses on theory and therefore we did not put much emphasis on the experimental evaluation, we still carefully design our experiments to test the robustness of our algorithm across different environments. Since our work is the first one to study the many-to-one setting with uniqueness conditions, there are indeed no comparable baselines. It is possible to design some sub-optimal algorithms in which each agent runs a MAB algorithm independently and there is no communication block among agents. However, such algorithm may not find the stable matching and thus suffers a linear regret.

**Optimality of our bound and the lower bound**    Recall that our bound is $O(NK\frac{\log(T)}{\Delta^2})$. There exists a lower bound of $O(\frac{\log(T)}{\Delta^2})$ under the setting where arms have the same and known preferences Sankararaman et al. (2021), which is a special case of our setting. Our bound is optimal in terms of $T$ and $\Delta$. For $N$, since each agent $j$ needs to face collisions from non-dominated arms and other agents, regret is bounded over the summation of agents and thus leads to the term $O(N)$. Usually in a multi-player decentralized setting Avner & Mannor (2014); Rosenski et al. (2016), each agent will suffer regret of term $N$ since it will be collided with other agents. Thus we conjecture such $N$ is unavoidable. For $K$, since in the decentralized setting, agents have no knowledge of arm preference, each agent needs to try each $O(\log(T)/\Delta^2)$ times to identify the stable matched arm. And it may get collided when pulling the other agent's stable matched arm, thus leading to the term $K$. $K$ might be removed for those agents who may never get collisions due to the special market structure.

---

[8]This assumption Roth & Sotomayor (1992); Akahoshi (2014); Altinok (2019) in our setting states that the addition of another agent $p_{i''}$ will not influence the preference ranking for an arm to agent $p_i$ and $p_{i'}$, i.e. $p_{i''} \cup p_i \succ_{a_j} p_{i''} \cup p_{i'}$ is equivalent to $p_{i'} \succ_{a_j} p_i$

### D.3 Strict Preference and "Indifferent Agents"

Our work focuses on strict preference rather than the more general case that considering indifferent agents. As far as we know, a lot of works studying the traditional (offline) matching markets would assume preferences to be strict Gale & Shapley (1962); Karpov (2019); Gutin et al. (2021); Nguyen et al. (2021); Akahoshi (2014), perhaps due to the reason of simplicity. Our work mainly follows these existing settings of the offline matching markets Gale & Shapley (1962); Karpov (2019); Gutin et al. (2021); Nguyen et al. (2021); Akahoshi (2014) and the bandit learning on the one-to-one matching markets Basu et al. (2021); Liu et al. (2020a); Sankararaman et al. (2021); Liu et al. (2020b) that assume strict preferences.

Note that if the agents are indifferent (or nearly indifferent) over the arms that are far down the ranking lists and do not affect the stable matching, our algorithm and analysis can actually go through. The gap appeared in the regret bound actually depends only on the those "(nearly) optimal" arms that appear in the stable matching or are the best among those not appeared in the stable matching.

Recall that our setting is to learn a particular stable matching, like previous works Basu et al. (2021); Liu et al. (2020a); Sankararaman et al. (2021); Liu et al. (2020b) learning the unique, or agent-pessimal/optimal stable matching on the one-to-one setting. Under this objective, if the agents are nearly indifferent, not exactly indifferent, over "(nearly) optimal" arms, no matter how small the gap is, the agents will need to figure out the which arm is better and the gap appears as the learning hardness. This phenomenon is common in multi-armed bandits where differentiating the optimal arm and the second optimal arm is the most difficult part of the learning. Then one might be curious about the objective to learn a "nearly stable matching". This would be more general and would prefer to leave it as interesting future work.

For the case when agents are exactly indifferent on "(nearly) optimal" arms, the stable matchings would not be unique. In this case, the communication block and the global deletion set of our algorithm need to be revised to allow each agent to keep more than one stable matched arm. Note that after this revision, the selected matching will not become fixed during interactions and will switch between all optimal stable matchings since the learning algorithm needs to continue exploring these arms to take precautions against the case of small gap. This will result in a phenomenon of fast-changing matching-selections, compared with our setting and most previous works Basu et al. (2021); Liu et al. (2020a); Sankararaman et al. (2021); Liu et al. (2020b) where the learning algorithm tends to stick on a specific matching in the latter learning period.

### D.4 Future Directions for Many-to-one Setting

First, we propose some interesting directions about the setting. This paper considers preference over individuals rather than agent sets. For example, when the first and fourth employees have cooperation experience and the second and third employees have no cooperation experience before, the company may prefer to recruit 1-st and 4-th together rather than 1-st, 2-nd or 2-nd, 3-rd. That is, $1, 4 \succ_k 2, 3$ may occur for arm $k$ and $1, 2, 3, 4 \in [N]$. Further research can also take this combination effect as the starting point. We assume that the preferences over agents for arms are known in our setting[9]. When multiple agents are accepted by one arm simultaneously, the ranking of these agents cannot be judged if under the assumption of unknown preference ranking. Therefore, the algorithm for rank estimation still needs further design. And our work is based on fixed finite agents set and arms set, thus how to generalize this setting to a dynamic one?

---

[9]The preference profile over arms for agents is unknown in our setting, and needed to be learned.

