# OpenReview forum: "Bandit Learning in Many-to-one Matching Markets with Uniqueness Conditions"
_ICLR.cc/2023/Conference — Submitted to ICLR 2023_

### Official Review · Reviewer_dQX3 · 2022-10-23

**Confidence:** 3
**Correctness:** 4
**Technical Novelty And Significance:** 2
**Empirical Novelty And Significance:** Not applicable
**Recommendation:** 5

**Clarity, Quality, Novelty And Reproducibility:**

The results appear technically sound (though I did not take a very careful look at the appendices).

There are quite a few typos, e.g., these two appear very early on:
1. The Thompson Sampling reference appears out of place in the Related Works section (line 5 thereof).
2. "Setting" section: Para 3 -- .... a matching satisfies individual rationality (not individually).

The communication block of the algorithm is unclear; I think this certainly is one aspect that can seriously benefit from an improved exposition.

Re the novelty of this work, please see the section on Strengths and Weaknesses.

**Strength And Weaknesses:**

While certainly a very interesting model to study, this paper seems very similar in style (including the model and even the title) to (https://dl.acm.org/doi/pdf/10.1145/3511808.3557248). In light of cited reference, the originality of this submission appears somewhat questionable. The only significance I see is the authors improving the $\mathcal{O}\left( \log^2 T \right)$ upper bound in the decentralized case (established in cited paper) to $\mathcal{O}\left( \log T \right)$; it is unclear whether the improvement is attributable to the new algorithm, communication block, or its analysis. While I do believe there is some technical merit to this submission, it would undoubtedly help the readers if the authors can compare their model, approach, and results with the cited reference. It would also help highlight their innovations and contributions (if any) vis-\`a-vis the cited paper.

I do understand the cited reference might not have been publicly available at the time of submission, however, I think it is imperative at this point that the authors address aforementioned concerns given the apparent non-trivial overlap between the two works.

**Summary Of The Paper:**

This paper studies an allocation problem where multiple "agents" can be assigned to the same "arm" in each round. Each arm has some capacity, and the combined capacity of the $K$ arms is large enough to accommodate all $N$ agents. Both sides of the market (agents and arms) have preferences over each other. Agent preferences over arms are unknown a priori, but can be learnt though sequential interactions with arms. If the number of agents bidding for the same arm (say, arm $k$) exceeds the arm's capacity, only the top $q_k$ (capacity of arm $k$) agents preferred by the arm are assigned to it; rest remain unmatched during that round. For each matched pair, a stochastic $[0,1]$-valued reward is realized that the concerned agent can learn from. The authors provide conditions for a unique stable "many-to-one" matching, and propose a decentralized algorithm (each agent independently runs its own thread of said algorithm) that guarantees a regret of $\mathcal{O}\left( \log T \right)$ for each agent relative to its match in the unique stable matching; $T$ being the number of matching rounds.

**Summary Of The Review:**

The overall quality of writing definitely warrants a minor revision. However, the biggest concern for me is the similarity with (https://dl.acm.org/doi/pdf/10.1145/3511808.3557248); the authors must appropriately address their contributions relative to this paper. Until that has been done, I will have my reservations about the merit/contribution of this submission.

---

> ### Author Response · Authors · 2022-11-17
> **Reply to Reviewer dQX3: Part 1**
>
> # About related work
>
> First, thanks for your comments. We will explain the differences between the work you mentioned [1] and this paper from the aspects of model assumptions, algorithm design, regret bound, etc.
>
> Although we have all analyzed the decentralized many-to-one market, this work is mainly carried out under the guarantee of uniqueness conditions. First, why we introduce uniqueness conditions: (i) The uniqueness condition encompasses many realistic preference structures including *Serial Dictatorship* and *No Crossing Conditions (NCC)* [2], mentioned in Section B.4. (ii) In the decentralized market, conflicts are unavoidable. We hope to reduce the conflicts between different agents through arm deletion, and the elimination process needs to ensure that the department of any subset of the market does not influence the stable matching. Intuitively, the uniqueness condition is to let both sides of the market reach an agreement to complete the matching, it is a reasonable condition. (iii) It is fairer to both sides that there would be no dispute about adopting stable matching preferred by which side. Note that the outcome of the GS algorithm would prefer the proposal side and would be unfair to the other side [1, 2]. (iv) The uniqueness condition induces a hierarchy in matching market in the process of arm deletion, hence reducing the regret bound derived by collision block, the main resource of the regret, to the number of matchings with sub-optimal arms in a induction method (in Appendix A.3). This makes the order of regret is $O(\frac{NK \log(T)}{\Delta^2})$, reaching the lower bound related to time horizon $T$ and reward gap $\Delta$ in the matching market with bandit algorithm [3]. There may be other conditions that reduce collisions in the decentralized market. However, the $\tilde{\alpha}$-condition proposed by us can ensure the control of regret order, and it is well implemented in our algorithms and experiments. And we give the proof that this condition is relatively weak (Section 4.1.2).
>
>
> Secondly, by adding the uniqueness condition and the algorithm design of arm deletion, the MO-UCB-D4 algorithm incurs a stable regret of $O(\frac{NK \log(T)}{\Delta^2})$, compared with the previous order $O(\frac{ \exp(N^4) N^5 K^2 \log^2(T)}{\Delta^2})$.
>
> In addition to this important improvement, the previous algorithm may fall into a loop, that is, a large regret occurs with a small probability, which can be seen in Example. They guarantee the order of regret in the average sense, and our algorithm will avoid their linear regret. Our algorithm can not only reduce the collision among agents, but also better identify the unknown preferences, which is also a good inspiration for the subsequent design and learning of the algorithm of unknown preferences.
>
> ### Example
> **(Loop in previous work)** As MOCA-UCB algorithm could cover one-to-one setting, we take one-to-one setting as an example.
> Consider the following preference profile (with $q_1 = 1, q_2 = 1$), the unique stable matching is $(s_1, c_1)$, $(s_2, c_2)$.
>
> | Arms      | Preference |
> | ----------- | ----------- |
> | $c_1:$    | $s_1 > s_2$     |
> | $c_2:$    | $s_2 > s_1 $    |
>
> | Agents      | Preference |
> | ----------- | ----------- |
> | $s_1:$    | $c_1 > c_2$     |
> | $s_2:$    | $c_1 > c_2 $    |
>
>
> Suppose both agent $s_1$ and agent $s_2$ implement MOCA-UCB with zero probability of delay. By a random initialization, it is possible that both them select arm $c_1$ at at first. Then, $s_2$ loses the conflict he will choose $c_2$ next round, which is the only arm in his plausible arm set. The UCB index of agent $s_1$ for arm $c_2$ is positive infinity at this point because they have not pulled it yet. Hence, $s_1$ attempts to pull $c_2$ at the second time step. Since $c_2$ prefers $s_2$, $s_1$ loses the conflict and his UCB index for arm $c_2$ remains infinite. The same argument shows that both agents will keep choosing the same arm, alternating between $c_1$ and $c_2$. As long as they stay in this cycle, both agents experience a constant stable regret.
>
> Hence, we give the comparisons between the work you mentioned and this work:
>
> ## References:
>
> [1] Zilong Wang, Liya Guo, Junming Yin, and Shuai Li. Bandit learning in many-to-one matching
> markets. In Proceedings of the 31st ACM International Conference on Information & Knowledge
> Management, pp. 2088–2097, 2022
>
> [2] Simon Clark. The uniqueness of stable matchings. Contributions in Theoretical Economics, 6(1),
> 2006
>
> [3] Abishek Sankararaman, Soumya Basu, and Karthik Abinav Sankararaman. Dominate or delete:
> Decentralized competing bandits in serial dictatorship. In International Conference on Artificial
> Intelligence and Statistics, pp. 1252–1260. PMLR, 2021

---

> > ### Author Response · Authors · 2022-11-17
> > **Reply to Reviewer dQX3: Part 2**
> >
> > (Continued from previous page)
> >
> > | Comparisons     | The work you mentioned  |   This work      |
> > | :---        |    :----:   |          :---: |
> > | Main Problem      | Many-to-one matching problem       | Many-to-one matching with uniqueness condition    |
> > | Setting     | Both centralized and decentralized         | The general case: decentralized      |
> > | Regret Bound    | $O(\frac{\exp(N^4) N^5 K^2 \log^2(T)}{\Delta^2})$       | $O(\frac{NK \log(T)}{\Delta^2})$: has an improvement    |
> > | Model  | Use a delay parameter to construct a new plausible set or to match the same arm with last step in each step    | With multi phases and arm deletion process to get full learning of unknown preferences and stable matching   |
> > |  Main Contributions  |  From one-to-one to many-to-one setting  |  1. From one-to-one to many-to-one setting; 2. Propose a new uniqueness condition and establish the relationship between this condition with other existing conditions; 3. Under the relatively weak $\tilde{\alpha}$-condition, our algorithm has a good performance, and the regret order reach the lower bound in terms of $T$ and $\Delta$ |

---

> > > ### Comment · Reviewer_dQX3 · 2022-11-24
> > > **Post author-rebuttal**
> > >
> > > I thank the authors for providing a detailed response. I do now see the differences between the two works, but I would like to see a concise discussion on this incorporated into the revision. I will keep my current score.

---

### Official Review · Reviewer_XTzP · 2022-10-23

**Confidence:** 5
**Correctness:** 4
**Technical Novelty And Significance:** 3
**Empirical Novelty And Significance:** 3
**Recommendation:** 6

**Clarity, Quality, Novelty And Reproducibility:**

Clarity

+ As I already mentioned above, the paper is clearly presented and the contributions are appropriately written without any over-selling.

Quality

+ The paper is reasonably high quality and the results are indeed significant. Bandits in matching market is an important line of work with many direct practical applications. Thus, extending these to as many canonical matching markets as possible is generally useful. This paper considers one such extension with many applications.

Novelty

+ The considered setting is novel and the proposed algorithm optimally solves this practical setting. I should note however that the algorithmic template itself is borrowed from prior work, but it still requires careful work to make it work in this new setting.

**Strength And Weaknesses:**

Strengths

+ The studied setting is important and practical for a wide range of applications. Many-to-one matching markets are very common in internet age companies and bandit learning these markets presents an interesting challenge that this paper considers

+ Overall the paper is well-written, places prior work in context and is careful in claiming contributions while also acknowledging which parts of prior work they build on. The algorithm is natural and they also address common questions one may have such as reducing this to 1-to-1 matching by making many copies.

+ The results are optimal, and they consider a very general setting of unique stable matching.

Weakness

+ The primary weakness I find in the paper is not going deeper on applications. Right now, they only briefly mention an application that seems contrived. I would like the authors to consider multiple applications and expand on them, even in the appendix. As written, the paper may come across as contrived. But in my opinion, many-to-one matching has many stronger applications; I list a couple here.

  - Pool riding in ride-share. This is an important application where we can match a driver to multiple riders. And the goal is to maximize the long-term platform utility.
- Slate ranking in recommender systems. Here a user can be matched to multiple content at a single request.


**Summary Of The Paper:**

This paper extends the problem of bandit in matching markets from a 1-to-1 matching to many to one matching. Many to one matching are an important family with rich applications in domains like crowd-sourcing and pool ride-share. They extend the setting of optimal regret in the case of unique stable matching for the 1-to-1 matching to many-to-one. To achieve this extension, they need to identify a number of key components. First, they need to identify the analogue of alpha-condition in the many-to-one matching world. Next, they need to learn to deal with collisions. And finally, they need to put these pieces together to prove the optimal regret bound. They design an algorithm that is similar in template to the prior work on 1-to-1 matching where there is a notion of global deletion and local deletion and priove theoretical guarantees for this problem. They also corroborate this with empirical simulations.

**Summary Of The Review:**

Overall, I believe that the pros outwieighs the cons. The considered paper is practical and the algorithm is a natural extension of UCBD4. Among the couple of follow-ups I suggested above, I would like to see the paper engage with the application bit more deeply. A section dedicated to a few applications and how the model in the current paper can be modelled for those applications will make the impact much more stronger. Additionally/aspirationally, I also wonder if the authors can find real-world datasets to run their algorithm on the above said applications which would definitely make this a stronger paper. From theory pov, I think this paper does a good job of straddling prior work contributions while also mentioning the novel aspects. Overall, I am leaning towards an accept.

---

> ### Author Response · Authors · 2022-11-17
> **Reply to Reviewer XTzP**
>
>
> # Multiple Applications of Many-to-one Setting
>
> First, thanks for your comments that the application is indeed a problem worth paying more attention to.
>
> As you mentioned, in addition to short-term recruitment, a car-hailing platform is a good example. Each passenger can choose the driver who meets his/her expectations on the platform. This preference is based on the distance between the driver and himself/herself, the type of car, the driver's service score, etc. When a driver receives multiple orders, he/she will select the most $q$ preferred passengers according to the distance between the passenger and himself/herself, the passenger's integrity record (the frequency of often canceling orders without reason) and other conditions. One driver can receive multiple orders at the same time (carpooling or several passengers form a short time, and multiple orders can be completed in a short time), and $q$ is the quota his car can accommodate.
>
> Another example can be seen in Slate ranking in recommendation systems or online book borrowing platform. If the number of books in a book is $q$, multiple readers want to borrow them at a certain time period. The platform selects the first $q$ users with relatively higher scores based on the user's score (the score is determined according to the user's credit rating: whether the books are returned on time, borrowing frequency, etc.). Each user can only borrow one book at a time. If the subscription is not successful, he can choose to subscribe to another book. In a short time, books that have been borrowed by other users will no longer appear in this user's waiting area (corresponding to that the agent will not select its own rejected arm).
>
> Besides, e-commerce platforms like eBay and Amazon, match consumers to goods can be seen as many-to-one matching platform. Similarly, online fashion retailers (e.g., Rent The Runway, Stitch Fix) match clients to clothing items. These online platforms that limited commodities have multiple demanders, and the demand is a staged, which can be regarded as our motivation.

---

### Official Review · Reviewer_xyU7 · 2022-10-24

**Confidence:** 3
**Correctness:** 3
**Technical Novelty And Significance:** 3
**Empirical Novelty And Significance:** 3
**Recommendation:** 5

**Clarity, Quality, Novelty And Reproducibility:**

The work is generally easy to follow, and it would be better if the weaknesses above can be overcome.

**Strength And Weaknesses:**

Strength:
1. The paper generally looks fine and easy to follow.
1. To explain the motivation of its formulation, this work describes a vivid scene with the companies and workers.

Weaknesses:
1. Is there a matching lower bound on the regret?
2. Although the assumption of the unique stable matching ensures the consistency, the author(s) may also mention what will be the challenge in a setting without unique stable matching.

**Summary Of The Paper:**

This work focuses on the many-to-one matching problem in the bandits, and the uniqueness condition is fundamental in their problem setup. The author(s) designs the MO-UCB-D4 algorithm and derives an upper bound. The paper also presents experimental results to convince the efficacy of the proposed algorithm.

**Summary Of The Review:**

This work focuses on the many-to-one matching problem in the bandits with the uniqueness condition is fundamental in their problem setup. The author(s) designs the MO-UCB-D4 algorithm and derives an upper bound. The paper also presents experimental results to convince the efficacy of the proposed algorithm.

==================

Thanks for the response. The gap between the upper and lower bounds is $NK$ as you explained, which indicates that the bounds are not sufficiently tight but I see your efforts. I would keep my score.

---

> ### Author Response · Authors · 2022-11-17
> **Reply to Reviewer xyU7**
>
> First, thanks for your comments, and we will answer them as follows.
>
> # Whether there is a matching to reach regret lower bound
>
> Thanks for your comments, and we discussed the optimality of our regret bound in Section D.
> Recall that our bound is $O(NK\frac{\log(T)}{\Delta^2})$. There exists a lower bound of $O(\frac{\log(T)}{\Delta^2})$ under the setting where arms have the same and known preferences [1], which is a special case of our setting.
>
> Since our preference assumption is weaker, $K$ and $N$ exist in our setting. Our bound is optimal in terms of $T$ and $\Delta$.
>
> For $N$, since each agent $j$ needs to face collisions from non-dominated arms and other agents, regret is bounded over the summation of agents and thus leads to the term $O(N)$. Usually in a multi-player decentralized setting [2, 3], each agent will suffer regret of term $N$ since it will be collided with other agents. Thus we conjecture such $N$ is unavoidable.
>
> For $K$, since in the decentralized setting, agents have no knowledge of arm preference, each agent needs to try each $O(\log (T)/\Delta^2)$ times to identify the stable matched arm. And it may get collided when pulling the other agent's stable matched arm, thus leading to the term $K$. $K$ might be removed for those agents who may never get collisions due to the special market structure.
>
> # The challenge in a setting without unique stable matching
>
> First, stability is indispensable that the goal of matching market is to find a stable matching. For uniqueness, our algorithm cannot be implemented without this constraint that it is based on arm deletion process. It requires there always exists a unique stable matching if an arbitrary subset of stable pairs are deleted out of the system. $\tilde{\alpha}$-condition (or, Unqc) provides protection for this problem.
>
> Second, for general markets, agents will collide with other agents when their preferences are unknown, which will bring great regret.
> The uniqueness condition induces a hierarchy in matching market in the process of arm deletion, hence reducing the regret bound derived by collision block, the main resource of the regret, to the number of matchings with sub-optimal arms in a induction method (in Appendix B.3). This makes the order of regret is $O(\frac{NK \log(T)}{\Delta^2})$, reaching the lower bound related to time horizon $T$ and reward gap $\Delta$ in the matching market with bandit algorithm [1]. If in a decentralized market without additional assumptions, the regret caused by the collision may meet the order of the $O(\frac{poly (N,K) \log^2(T)}{\Delta^2})$.
>
> ## References
>
> [1] Abishek Sankararaman, Soumya Basu, and Karthik Abinav Sankararaman. Dominate or delete:
> Decentralized competing bandits in serial dictatorship. In International Conference on Artificial
> Intelligence and Statistics, pp. 1252–1260. PMLR, 2021
>
> [2] Orly Avner and Shie Mannor. Concurrent bandits and cognitive radio networks. In Joint European
> Conference on Machine Learning and Knowledge Discovery in Databases, pp. 66–81. Springer,
> 2014
>
> [3] Jonathan Rosenski, Ohad Shamir, and Liran Szlak. Multi-player bandits–a musical chairs approach.
> In International Conference on Machine Learning, pp. 155–163. PMLR, 2016

---

### Official Review · Reviewer_mBKs · 2022-10-25

**Confidence:** 4
**Correctness:** 4
**Technical Novelty And Significance:** 2
**Empirical Novelty And Significance:** 2
**Recommendation:** 5

**Clarity, Quality, Novelty And Reproducibility:**

The paper builds on an emerging line of work that views learning stable matchings as a multi-armed bandit problem, and considers the interesting setting of many-to-one matchings. However, the assumptions on the preferences made in the paper (see Weaknesses section) are very strong and seem to miss some of the interesting structure arising in many-to-one matching markets. Thus, the contribution is somewhat incremental over existing work on learning one-to-one stable matchings in a decentralized setting.

The paper is slightly difficult to follow. First, the notation in the model section (Section 2) is a bit messy and could be cleaned up. Moreover, it would be better to state preference assumptions (e.g. that the arm preferences are not over general sets of agents) in Section 2 rather than to mentioning these assumptions until Section 5. Lastly, the statement of the regret bound (Theorem 3) is rather messy.

**Strength And Weaknesses:**

Strengths:
- The problem of studying learning stable matchings in the many-to-one setting is well-motivated and timely, given that the emerging line on work learning stable matchings focuses on the one-to-one setting.
- The paper leverages uniqueness of stable matchings to achieve O(log T) regret.

Weaknesses:
- The implicit assumption on the preferences of the arms—that arm preferences decompose into a preference ranking over individual agents—is very strong. Much of the interesting structure in many-to-many matching markets comes from arm preferences being over sets of agents. In particular, preferences might depend on the interactions between the agents within the set (e.g. the arms might prefer “balanced” sets of agents). The assumption made in this paper disallows these types of interactive structures. The model thus only applies to matching markets with capacity constraints, and does not apply to general many-to-one matching markets.
- While the regret bound is shown to be have optimal dependence on T and Delta (see Section C.2), the paper does not show optimality of the dependence on N and K.
- One weakness of the regret notion studied in this paper—stable regret—is that it is discontinuous in the utilities of agents: this makes it impossible to achieve instance-independent regret bounds. As a result, the paper only provides an instance-dependent regret bound and likely cannot achieve instance-independent regret bounds (similar to prior work on decentralized learning in one-to-one matching markets). It would be helpful to discuss this limitation of the model and regret definition.
- Some references to related work are missing, e.g. Das and Kamenica (IJCAI 2005), Jagadeesan, Wei, Wang, Jordan, and Steinhardt (NeurIPS 2021), Cen and Shah (AISTATS 2022), Min, Wang, Xu, Wang, Jordan, Wang (NeurIPS 2022).


**Summary Of The Paper:**


The paper investigates learning stable matchings in many-to-one matching markets where agents (“arms”) on one side can be matched to sets of agents on the other side. Building on an emerging line of work, the paper views the problem of learning stable matchings within the multi-armed bandits framework. The paper considers and analyzes a condition that guarantees uniqueness of stable matchings. Under this uniqueness condition, the paper designs a bandit algorithm that achieves O(log T / delta^2) "stable regret". The algorithm proceeds in phases, where each phase includes a regret minimization block and a communication block. The paper validates the performance of the algorithm through simulations.

**Summary Of The Review:**

The paper studies the timely problem of learning stable matchings in many-to-one matching markets. However, the preference assumptions are very strong and, as a result, the technical contribution is somewhat limited.

---

> ### Author Response · Authors · 2022-11-17
> **Reply to Reviewer mBKs: Part 1**
>
> First of all, thanks for your comments, and we will answer as follows.
>
> # Only the contribution from one-to-one to many-to-one is weak
>
> In addition to extending the traditional one-to-one setting to the many-to-one problem, the contributions of this paper are also worth mentioning the following points: First, the $\tilde{\alpha}$-condition is a new uniqueness condition we proposed, and its corresponding properties and the feasibility of the algorithm under this condition are given. Secondly, the work establishes the relationships between the existing many-to-one uniqueness conditions to ensure the wide applicability of the weaker $\tilde{\alpha}$-condition.
>
>
> As mentioned in the previous work [1], the lower bound of the algorithm in the matching market is $O(\frac{ \log(T)}{\Delta^2})$, Our algorithm has reached lower regret bound related to time horizon $T$ and reward gap $\Delta$ in many-to-one market. And their one-to-one setting is a special case of our setting that many-to-one matching need to consider more. Hence, our regret has reached its optimality.
>
>
> In addition, the transition from one-to-one to many-to-one is not smooth. Even if our preferences are specific to individuals, it does not mean that the different replicas that be accepted by one arm are $q$ independent individuals with the same preference, and there is competition between them. When multiple agents choose the same arm $k$, as he can accept more than one agent, these agents still cannot judge who is more preferred by arm $k$, which makes it difficult to identify the dominated arms in the communication block, and it also takes more time to form the stable matching, hindering the control of the regret order.
>
> # Arm preference over individuals is too strong
>
> Inspired by previous work [2, 3, 4], we consider the preference over individuals rather than groups. Based on this assumption, we can assume that the arm preference ranking for different agents is strict, which is difficult to guarantee in group preference. Then the reward gap $\Delta$ in regret bound is strictly greater than $0$, making regret not infinite. In addition, preference over sets or group makes regret order increase exponentially with respect to $N$, $K$, which bring much regret. And this assumption has certain practicability, such as the model that there is almost only competition between agents such as school enrollment and house leasing [5, 6].
>
> # Optimality of the dependence on $N$ and $K$ in regret bound
>
> As we analyzed in Section D, our bound $O(NK\frac{\log(T)}{\Delta^2})$ is optimal in terms of $T$ and $\Delta$. Although there exists a lower bound of $O(\frac{\log(T)}{\Delta^2})$ under the setting where arms have the same and known preferences [1], which is a special case of our setting.
>
> For $N$, since each agent $j$ needs to face collisions from non-dominated arms and other agents, regret is bounded over the summation of agents and thus leads to the term $O(N)$.
> Usually in a multi-player decentralized setting [7, 8], each agent will suffer regret of term $N$ since it will be collided with other agents. Thus we conjecture such $N$ is unavoidable.
>
> For $K$, since in the decentralized setting, agents have no knowledge of arm preference, each agent needs to try each $O(\log (T)/\Delta^2)$ times to identify the stable matched arm. And it may get collided when pulling the other agent's stable matched arm, thus leading to the term $K$. $K$ might be removed for those agents who may never get collisions due to the special market structure.
>
>
> ## References
>
> [1] Abishek Sankararaman, Soumya Basu, and Karthik Abinav Sankararaman. Dominate or delete:
> Decentralized competing bandits in serial dictatorship. In International Conference on Artificial
> Intelligence and Statistics, pp. 1252–1260. PMLR, 2021
>
> [2] Alvin E Roth and Marilda Sotomayor. Two-sided matching. Handbook of game theory with economic
> applications, 1:485–541, 1992
>
> [3] Jay Sethuraman, Chung-Piaw Teo, Liwen Qian, et al. Many-to-one stable matching: Geometry and
> fairness. Mathematics of Operations Research, 31(3):581–596, 2006
>
> [4] Ahmet Altinok. Dynamic many-to-one matching. Available at SSRN 3526522, 2019.
>
> [5] David Gale and Lloyd S Shapley. College admissions and the stability of marriage. The American
> Mathematical Monthly, 69(1):9–15, 1962
>
> [6] Alvin E Roth. The college admissions problem is not equivalent to the marriage problem. Journal of
> economic Theory, 36(2):277–288, 1985
>
> [7] Orly Avner and Shie Mannor. Concurrent bandits and cognitive radio networks. In Joint European
> Conference on Machine Learning and Knowledge Discovery in Databases, pp. 66–81. Springer,
> 2014
>
> [8] Jonathan Rosenski, Ohad Shamir, and Liran Szlak. Multi-player bandits–a musical chairs approach.
> In International Conference on Machine Learning, pp. 155–163. PMLR, 2016

---

> ### Author Response · Authors · 2022-11-17
> **Reply to Reviewer mBKs: Part 2**
>
> # Our instance-dependent regret bound
>
> First, a lot of works both studying the traditional (offline) matching markets [5, 9, 10, 11, 12] and online one-to-one market of bandit setting [1, 13, 14, 15] would be based on strict preferences rather than continuous utilities, perhaps due to the reason of simplicity. The discontinuous preference is easy to explain the matching rules in real life.
>
>
> Second, as you mentioned that it is impossible for discontinuous preference to achieve instance-independent regret bounds. We then illustrate the rationality our instance-dependence regret. Although our regret bound depends on reward gap $\Delta$, we focus on strict preference that $\Delta > 0$ always holds, hence the regret cannot reach infinity. According to the source of our reward gap $\Delta$, it is caused by the number of times agent $j$ and sub-optimal arm $k$ ($k$ is not a dominated arm) have matched with. As long as the adjacent preferences are close, it will bring difficulties to identify optimal arm and sub-optimal arm, so the regret will very large, or the regret are vulnerable to minor changes in preferences, resulting in large disturbances, i.e. dependent on instance. However, it is unavoidable for the model based setting without additional restrictions on preferences. As proved in some examples with instance-independent regret [16], they learn the minimum norm of the difference between the $\theta_{T_i}$ in stage $T_i$ and the optimal $\theta^{\ast}$ through adaptive algorithm, and can get rid of the control of instance-dependent term; However, if we also apply the restriction that the minimum reward gap $\Delta$ is greater than a small positive number according to the preference sequence, we can also get an order independent of $\Delta$, the instance-dependent term in our regret bound.
>
>
> In fact, without more restriction, our regret bound is reasonable that $\Delta > 0$ strictly. Although for the case that an arm has similar preferences for two agents, it do not affect the stable matching, our algorithm and analysis can actually go through. The gap appeared in the regret bound actually depends only on those ``(nearly) optimal'' arms that appear in the stable matching or are the best among those not appeared in the stable matching.
> Recall that our setting is to learn a particular stable matching, like previous works [1, 13, 14 15] learning the unique, or agent-pessimal/optimal stable matching on the one-to-one setting.
>
>
> After your comments, it is indeed a worthwhile direction to use utility functions to depict preferences. We will also add improvements to this problem in the subsequent work.
>
> ## References
>
> [9] Alexander Karpov. A necessary and sufficient condition for uniqueness consistency in the stable
> marriage matching problem. Economics Letters, 178:63–65, 2019
>
> [10] Gregory Z Gutin, Philip R Neary, and Anders Yeo. Unique stable matchings. arXiv preprint
> arXiv:2106.12977, 2021
>
> [11] Hai Nguyen, Thành Nguyen, and Alexander Teytelboym. Stability in matching markets with complex
> constraints. Management Science, 67(12):7438–7454, 2021
>
> [12] Takashi Akahoshi. Singleton core in many-to-one matching problems. Mathematical Social Sciences,
> 72:7–13, 2014
>
> [13] Soumya Basu, Karthik Abinav Sankararaman, and Abishek Sankararaman. Beyond log2(t) regret for
> decentralized bandits in matching markets. In International Conference on Machine Learning, pp.
> 705–715, 2021
>
> [14] Lydia T Liu, Horia Mania, and Michael Jordan. Competing bandits in matching markets. In
> International Conference on Artificial Intelligence and Statistics, pp. 1618–1628. PMLR, 2020
>
> [15] Lydia T Liu, Feng Ruan, Horia Mania, and Michael I Jordan. Bandit learning in decentralized
> matching markets. arXiv preprint arXiv:2012.07348, 2020b
>
> [16] Avishek Ghosh and Abishek Sankararaman. Breaking the \√{T } barrier: Instance-independent
> logarithmic regret in stochastic contextual linear bandits. arXiv preprint arXiv:2205.09899, 2022

---

### Decision · Program_Chairs · 2023-01-20

**Decision:**

Reject

**Justification For Why Not Higher Score:**

The contributions are actually a bit slim wrt to the existing literature.

**Justification For Why Not Lower Score:**

N/A

**Metareview: Summary, Strengths And Weaknesses:**

This paper is part of the now growing trend of literature of learning (stable) matching in bi-partite market. The contribution of this paper is to consider many-to-one instead of one-to-one matching.

This paper is rather well written, clear and easy to follow, but the novelties are rather slim compare to the existing literature; of course, it improves upon existing algorithms in several directions, but without reaching optimal (with respect to all parameters) at the cost at rather strong assumption.

The referees were also rather lukewarm, comforting my personal opinion of that paper. I think it is a good paper, but unfortunately, it does not reach the bar for ICLR this year.